

# Scientific Workflows Applied to the Coupling of a Continuum (Elmer v8.3) and a Discrete Element (HiDEM v1.0) Ice Dynamic Model

Shahbaz Memon[1,4], Dorothée Vallot[2], Thomas Zwinger[3], Jan Åström[3], Helmut Neukirchen[4], Morris Riedel[1,4], and Matthias Book[4]

[1]Jülich Supercomputing Centre, Forschungszentrum Jülich, Leo-Brandt Straße, 52428 Jülich, Germany
[2]Department of Earth Sciences, Uppsala University, Uppsala, Sweden
[3]CSC – IT Center for Science Ltd., Espoo, Finland
[4]Faculty of Industrial Engineering, Mechanical Engineering and Computer Science, University of Iceland

**Correspondence:** Shahbaz Memon (m.memon@fz-juelich.de)

**Abstract.** Scientific computing applications involving complex simulations and data-intensive processing are often composed of multiple tasks forming a workflow of computing jobs. Scientific communities running such applications on distributed and heterogeneous computing resources find it cumbersome to manage and monitor the execution of these tasks. Scientific workflow management systems (WMS) can be used to automate and simplify complex task structures by providing tooling for the

composition and execution of workflows across distributed and heterogeneous computing environments. As a case study, we apply the UNICORE workflow management system to a formerly hard-coded coupling of a glacier sliding and calving simulation that contains many tasks and dependencies, ranging from pre-processing and data management to repetitive executions in heterogeneous high-performance computing (HPC) resource environments. Using the UNICORE workflow management system, the composition, management, and execution of the glacier modelling workflow becomes easier with respect to usage,

monitoring, maintenance, re-usability, portability, and reproducibility in different environments and by different user groups.

## 1 Introduction

The complexity of glaciological systems is in an increasing way reflected by the physical models used to describe the processes acting on different temporal and spatial scales. Addressing those complexities inevitably involves the combination of different sub-models into a single simulation that encompasses multiple tasks executed in a distributed computing facility. A particularly

good example for such a combination is the simulation of calving behaviour at the front of glaciers that combines continuum model and discrete element model simulations. These computational tasks are connected to each other to form a *scientific workflow:* the composition and execution of multiple data processing steps as defined by the requirements of the concerned scientific application.

  Carrying out the discrete calving and ice flow modelling as one workflow instance becomes very laborious without an

automated mechanism. An analysis of such a workflow in more detail reveals that there are even more steps than just the two model elements mentioned above: These include, for instance, pre- and post-processing, job dependency management, job



submission, monitoring, conditional invocations, and multi-site data management. These steps can be cumbersome for a user, in particular if any step may produce an unexpected output, which can cause the whole workflow to fail, and may thus require to re-run the whole workflow. A *Workflow Management System* (WMS) allows to automate and ease these steps by means of abstraction, which not only increases usability, but also enhances portability to and reproducibility on different computing
platforms.

The main contribution of this article is to identify the workflow problems to be solved for coupling a glacier continuum model and a discrete element model, to elicit corresponding requirements, and to implement an automated workflow based on the UNICORE (Streit et al., 2010) distributed computing middleware, in particular using the UNICORE workflow management system (Memon et al., 2007). We demonstrate this by taking a shell script that contains a hard-coded low-level workflow and
turning it into a high-level easy-to-use scientific workflow.

This article is structured as follows: Subsequent to this introduction, foundations on glacier calving modelling and on scientific workflows are provided in Section 2. The targeted glacier modelling use case is presented in Section 3 which describes a workflow baseline used for model coupling and the applications used for execution. Afterwards, in Section 4, the problems of this initial workflow are discussed. It is then demonstrated how to create a better solution, by first identifying requirements to
solve these problems (Section 5), followed by creating an improved matching workflow design (Section 6), and finally implementing the workflow (Section 7). The implemented workflow is evaluated in Section 8 from different perspectives, including a discussion of how the identified requirements from Section 5 have been fulfilled. Finally, a summary and an outlook conclude the article.

## 2   Foundations

### 20  2.1   Modelling and Simulating the Calving of a Glacier

The calving behaviour at the front of glaciers is still a largely unresolved topic in modern theoretical glaciology. The core of the problem is that ice as a material shows different behaviour, depending on the time scale on which the forces are applied (Greve and Blatter, 2009). The everyday experience is that ice is a brittle solid body (e.g. breaking icicles). Such behaviour is observed if the reaction to a force is in the range of seconds to minutes. Theoretical glaciology up to recent years rather dealt with the
long-term (i.e., beyond minutes to millennia) behaviour of glaciers and ice sheets, where ice shows the property of a strong non-linear, shear thinning fluid (Greve and Blatter, 2009). This leads to a description of long-term ice flow dynamics in classical ice-sheet and glacier dynamics models (Gagliardini et al., 2013) in terms of thermo-mechanically coupled non-Newtonian Stokes flow continuum models.

In stark contrast to such a description, the process of cracking or calving (i.e., the complete failure of ice fronts) is an
inherently discontinuous process, that – if addressed in a physically correct way – needs a completely different model approach. Models adopting a discontinuous approach have been developed throughout the recent years (e.g. Åström et al., 2013; Bassis and Jacobs, 2013). These models describe the glacier as discrete particles connected by elastic beams that can be dissolved if a certain critical strain is exceeded, thus being able to mimic elastic as well as brittle behaviour of ice. The size of these model



particles (in the range of metres) which need to resolve a whole glacier of several cubic kilometres inherently demands large computational resources. In addition, the characteristic speeds of advancing cracks is close to the speed of sound, which, in combination with the small spatial resolution, imposes a maximum allowed timestep size of a fraction of a second.

In other words, the combination of a discrete element and a continuum ice flow model is a temporal multi-scale problem, where the first basically describes an instant change of geometry or rheology for the latter. This means that both models need to be run in a sequential manner, with several repetitions. This defines a workflow of two model components that strongly differ in computational demand. In addition, these two model components have to effectively and efficiently exchange data – namely, the new geometries either changed by flow deformation or by calving as well as damage caused by fraction.

## 2.2   Scientific Workflows

A scientific workflow can be defined as the composition and execution of multiple data processing steps as required by a scientific computing application, i.e. e-Science. Such a workflow captures a series of analytical steps of computational experiments to "aid the scientific discovery process through the combination of scientific data management, analysis, simulation, and visualisation" (Barker and van Hemert, 2008). Conceptually, scientific workflows can be considered (and are typically visualized) as graphs consisting of nodes representing individual tasks and constructs, and edges representing different types 15  of associations between nodes, such as sequential or conditional execution.

    Carrying out the discrete calving and ice flow model simulations becomes very complex as two (or even more) parallel *High-Performance Computing* (HPC) applications are involved, especially if there are tasks in the workflow that consist of pre- and post-processing phases, and require multi-site and iterative job and data management functions. The overall scenario may easily become unmanageable, and the workflow management might be prone to errors and failures.

Such a workflow scenario will be even more challenging when some parts are launched on heterogeneous resource management systems equipped with different file systems, different data transfer mechanisms, and different job submission systems. In our case, for example, two different HPC clusters with different characteristics are simultaneously used, one for the ice flow modelling and another one for the discrete element (i.e., calving) modelling executions.

    These workflow challenges can be addressed by a *Workflow Management System* (WMS) that is capable of managing the 25  complex dependencies of many job steps with multiple conditional and nested constructs. Several scientific WMSs have been developed to automate complex applications from multi-disciplinary backgrounds. Ferreira da Silva et al. (2017) comprehensively categorized different workflow management systems according to the type of execution scenario and capabilities they offer, and also identified their specialised scientific domain. Several WMSs have been developed and are widely used to support complex scientific scenarios, among them Pegasus (Deelman et al., 2015), Kepler (Ludäscher et al., 2006) and Taverna (Oinn 30  et al., 2006), to name just a few. These WMSs are also used for specialised scenarios and cater to a specific group of users who interact with HPC resources.

    Our scenario from the domain of glaciology using HPC technology includes a complex workflow graph, and is therefore not easily manageable for users having only little expertise concerning HPC environments. Hence it is important to implement the glacier modelling use case with a WMS that offers a rich graphical front-end and simultaneously offers seamless





execution of involved applications. Considering that, the glacier coupling and calving use case is automated through our standards-based workflow management system (Memon et al., 2007) that is part of the *Uniform Interface to Computing Resources* (UNICORE) (Streit et al., 2010) distributed computing middleware. It is specifically designed to support HPC applications deployed in a massively parallel environment. As described later in Section 7, our WMS for UNICORE provides a rich

graphical interface for the composition, management and monitoring of scientific workflows by users with different levels of system expertise.

## 3 Use Case: Kronebreen Glacier Simulation

Coupling a continuum ice flow model and a particle-based calving model of the Kronebreen glacier provides a well-suited case study for the application of a WMS. In the following, we describe the involved software executables and the underlying

workflow. The challenge is that this workflow was only available as a single shell script that is hard to understand with many hard-coded aspects.

### 3.1 Conceptual Scheme

Kronebreen is a tidewater glacier (ice flowing directly into the ocean) that is one of the fastest-flowing glaciers of the Svalbard archipelago. After a period of stability, it started to retreat in 2011 and continued since then. This glacier has been largely

studied (e.g. Kääb et al., 2005; Luckman et al., 2015; Nuth et al., 2012; van Pelt and Kohler, 2015; Schellenberger et al., 2015), partly due to its situation close to a research station and its interesting behaviour in terms of sliding and calving. For that reason, it is a good candidate for the present study. The aim is to reproduce both continuous (ice flow) and discrete (calving) processes using a *Finite Element Model* (FEM) and a first-principle ice fracture model, respectively.

### 3.2 Applications: Meshing Tools, Continuum and Discrete Ice Dynamics Model

Within our application, we use the continuum model Elmer/Ice (Gagliardini et al., 2013), and for the discrete calving model, the *Helsinki Discrete Element Model* (HiDEM) (Åström et al., 2013). Both codes can be run on large parallel HPC platforms. Implied by the physics that have to be addressed and by the modelling tools, the workflow contains three main applications that are part of the use case implementation:

1. **Meshing: Gmsh** (Geuzaine and Remacle, 2009) is the applied meshing tool that provides the later-on extruded footprint
mesh for the continuum ice flow model run. Gmsh is an open source versatile and scriptable meshing tool that perfectly matches the demands of being deployed within a workflow like this one. Gmsh applies a bottom-up geometry and meshing strategy, starting from outline points of the glacier, building a closed loop of its outline and further creating a planar surface that is being meshed in two dimensions.

2. **Continuum Modelling: Elmer/Ice** (Gagliardini et al., 2013) is the open source ice sheet model used to compute the
long-time dynamics of the glacier. Elmer/Ice is based on the multi-physics package Elmer (Råback et al., 2018), an open



source Finite Element code developed by CSC – IT Center for Science Ltd. Elmer/Ice is able to utilize parallel processing, applying the *Message Passing Interface* (MPI) paradigm (Message Passing Interface Forum) that uses messages to exchange data between nodes of a distributed parallel processing environment, and – for certain solver implementations – OpenMP (Dagum and Menon, 1998) for shared-memory parallel processing using multi-threading. Elmer also provides

the `ElmerGrid` executable that can be used to convert the mesh created by Gmsh and at the same time perform a domain decomposition on the footprint, using the Metis library. The solver executable `ElmerSolver` has a built-in feature to internally extrude the given footprint mesh into layered columns of prisms and impose the given surface and bedrock elevation to form the volume of the glacier. Elmer is built on a shared library concept, meaning all solver modules are loaded during runtime. This enables the easy deployment of user-written functions and solvers though an API.

3. **Discrete Modelling: HiDEM** (Helsinki Discrete Element Model) (Åström et al., 2013) is a discrete element model that represents the glacier as mass-points connected by massless beams that are allowed to break if exceeding a given threshold. An additionally applied repelling potential based on distance guarantees a non-intersection of compressed loose particles. Solving Newton's equation on such a setup, it is possible to realistically reproduce the behaviour of fracturing. The downside of this approach is the high demand of computational power, as the several cubic kilometres

large glacier is discretised in pieces of a few tens of cubic metres. Furthermore, the time scales for the simulation are imposed by the ratio of the speed of sound to the typical length of the discretised particles, which falls clearly below seconds. Hence, despite the fact that the code is utilizing massive parallel computing using the MPI paradigm, only a few minutes to hours of physical time can be computed even on a huge HPC cluster. HiDEM receives the initial geometry from Elmer/Ice, in form of gridded data over a limited area at the tongue of the glacier, and also receives the basal friction

coefficient distribution computed within the continuum ice flow model.

### 3.3 Initial Base Workflow

The continuum ice flow model and the discrete calving model need to be coupled, thus leading to a scientific workflow. A baseline version of the workflow was initially realised by a Bash shell script that calls the above executables as well as additional Python helper code, and performs all the needed management operations using shell script commands. The underlying

workflow is as follows:

**Step 1: Generate the Mesh for Elmer/Ice**

At $t = t_i$, the front position $F_i$ and the contour $Cont$ are given as input to create the mesh $M_i$. This determines the domain of the glacier for the ice flow model Elmer/Ice. A Python script invokes the meshing executable `gmsh` to build the mesh from the contour and the front position, and invokes ElmerGrid (Råback, 2015) to convert it into Elmer format. In this particular

application, the mesh is then split into 16 partitions. Figure 1 provides an example of the mesh generated in the case of our Kronebreen case study. This step runs as a single-threaded application, as with respect to resource requirements, mesh generation is not very CPU-intensive (serial, i.e. 1 CPU core), but it consumes some storage space.





**Figure 1.** Mesh example of Kronebreen. Colors represent the surface elevation.

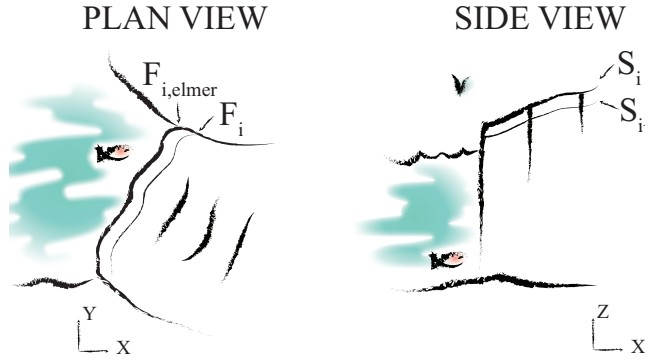

**Figure 2.** Conceptual plan and side view of an Elmer/Ice transient run. The initial front $F_i$ and surface $S_i$ are evolved to a new position of the front $F_{i,elmer}$ and a new surface elevation $S_{i+1}$.

**Step 2: Ice Flow Modelling and Conversion to HiDEM Domain**

The continuum ice flow model is executed using the `ElmerSolver` application which is an MPI-based implementation and part of the Elmer application suite. The number of time steps ($N_t$) depends on the studied glacier and process as well as the spatial resolution. Here, we simulate $N_t = 11$ days in one run, which – using a time-step size of one day – corresponds to the update frequency of remote sensing satellite data which was used for model validation.

As the basal boundary condition (BC), we assume a no-penetration condition with no basal melting nor accumulation and a basal friction law of any type (currently a Weertman friction law). The upper BC is defined as a stress-free surface and is able to evolve during the simulation following an advection equation forced by a surface mass balance. As the BC in contact with the ocean, the normal component of the stress vector is equal to the hydrostatic water pressure exerted by the ocean where ice is below sea level. The front is also able to evolve, using a Lagrangian scheme (i.e., mesh velocity equal to ice flow





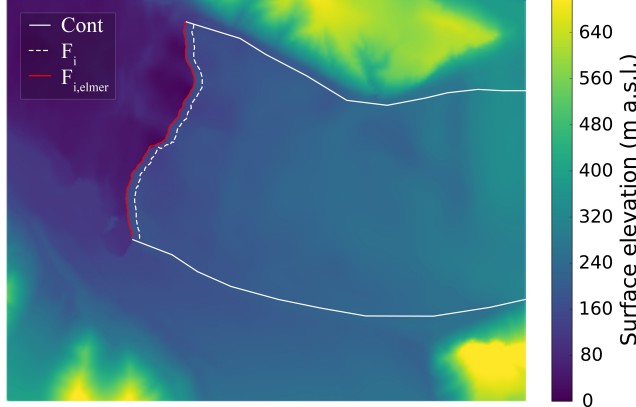

**Figure 3.** Surface elevation for the HiDEM (minimum bed elevation equal to 0), initial contour $C_i$ and new position of the front $F_{i,elmer}$.

velocity corrected by melting). The temperature profile in the ice and lateral BC are prescribed but can be changed easily. The ice is flowing as a non-linear isotropic viscous fluid following Glen's flow law (Cuffey and Paterson, 2010) and the Stokes equations for an incompressible fluid are solved over the ice volume. Elmer/Ice executes a solver input file, $\texttt{SIF}_i$, with the above-mentioned parameters.

After the simulation, the glacier has a new position of the front $F_{i,elmer}$, a new surface elevation $S_{i+1}$ (see Fig. 2) and a map of basal friction coefficients (determined by a linear sliding law). These form the output of Elmer/Ice and input to HiDEM. Elmer/Ice can be sped up by parallel processing, but is not as CPU-intensive as HiDEM, hence only 16 CPU cores are used for a relatively small glacier, like Kronebreen, in this part of the workflow.

The output format of Elmer/Ice does not match the input format of HiDEM, and HiDEM does not need to process the
whole glacier, but only the front that is relevant for calving. Hence, a conversion step runs a set of helper scripts ("Elmer to HiDEM") implemented in Python. This step is performing a conversion of the output of $\texttt{ElmerSolver}$ to the HiDEM grid ($10\,\mathrm{m} \times 10\,\mathrm{m}$ in our case) that is used for the calving front of the glacier. Elevations are offset so that the minimum bed elevation is equal to 0. It also includes places with no ice where the surface elevation is equal to the bed elevation (see Fig. 3). This conversion step creates a text input file for HiDEM, $Pin_i$, with coordinates, surface, bed and basal friction coefficient.
This conversion is performed on a single CPU.

**Step 3: Discrete Particle Modelling and Conversion to Elmer/Ice Domain**

The converted output from Elmer/Ice is passed to HiDEM. The implementation of HiDEM requires an MPI environment (560 cores running the MPI processes in our case), because in comparison to the other steps, it consumes the biggest share of CPU hours due to the high spatial and temporal resolution. In order to avoid excess computations, HiDEM scales down the
obtained friction parameters it receives from Elmer/Ice (in our case using the factor $10^{-4}$) so as to increase the sliding speeds and thereby reduce the physical time (in our case $100\,\mathrm{s}$) needed to evaluate the resulting fractures. This can be justified because the distribution of the friction parameters and hence the instantaneous fracture pattern does not change, and the time scales of



ice flow (represented by the continuum model) and fracturing are separate anyhow. A new front position, $F_{i+1}$, is modelled after the simulation. Because of the high spatial resolution, the involved text files are rather large.

Once a HiDEM run is completed, the next step is to re-convert this data set to the Elmer/Ice format, so that the next iteration of the coupled glaciology models' workflow can begin. Again, a Python script ("HiDEM to Elmer") is used to convert the

HiDEM output into a format that can be read by Gmsh and ElmerGrid. For this purpose, the new front position $F_{i+1}$ (after calving) and surface elevation $S_{i+1}$ are re-introduced into Step 1 at $t = t_{i+1}$. Again, the conversion is performed in serial execution. After this step, the workflow is begins again, starting from Step 1.

## 4    Issues of Initial Workflow

While the above workflow and its implementation using a shell script worked, both its design and its implementation had a

couple of issues that are discussed in the following.

The initial version of the workflow was implemented using a single shell script called `simu_coupling.sh` that contained more than 400 lines of code and invoked multiple applications. This single script encompassed all the workflow execution steps and monitoring assignments on a low level, making any extension and debugging extremely difficult for the script's author and almost impossible for anyone not familiar with the shell script.

As the complete coupling and calving workflow is implemented inside `simu_coupling.sh`, all the workflow-related multi-job iterations and error handling were realised in this central script. The individual tasks of the workflow, either parametric or atomic, incarnated as separate jobs in the used HPC batch system. Identifying failures or troubleshooting on any of the running and completed jobs required significant efforts.

The `simu_coupling.sh` script supported only a limited set of computational resources, i.e. it was low-level hard-coded

to the particular HPC cluster on which the implementation was intended to run. For example, the type of parallel execution environments, such as the MPI implementation to be used, were statically defined. Because of this, any updates of the dependent environment at the site level required changes at various locations in the shell script. Listing 1 shows part of the `simu_coupling.sh` script: Lines 12–14 contain the manual job command script specification, and lines 25–34 load required environment modules. Having the HPC environment set in very static manner in the form of job submission command

scripts and the related environment settings, as shown here, was a limitation: The main `simu_coupling.sh` script and the accompanying Python helper routines could not be easily ported to another computation environment that very likely has, e.g., a different HPC batch job submission system or file system layout. This resulted in a low degree of reproducibility due to a lack of portability: If other scientists wanted to take the files and run the application on a different resource environment or perform the simulation using a different data set, such a hard-coded workflow implementation would have to be adjusted

significantly, which requires high effort and bears a high risk of introducing bugs. Therefore, missing portability together with bad readability and low flexibility inhibited scientific reproducibility and re-use.

Another problem was inefficient resource usage: The script-based workflow implementation performed two data conversion tasks in every iteration, namely the previously mentioned Python scripts "Elmer to HiDEM" and "HiDEM to Elmer". These





```
nb_array=`awk '{print $1}' n_list.txt`
...
# Loop over runs
for nb in $nb_array
do
echo "**********************"
# Run the MeshToParticle.py
mkdir -p $outputfolder/Python_particle_n/Date$nb
echo '#!/bin/csh
#SBATCH -J MeshToElmer_'$nb'
#SBATCH -e t_MeshToElmer_'$nb'.err
#SBATCH -o t_MeshToElmer_'$nb'.log
#SBATCH --mem-per-cpu='$usemem'
##SBATCH -n '$noproc'
#SBATCH -t '$usetime1' ' > ${outputfolder}/Python_particle_n/Date${nb}/MeshToElmer_${nb}.sh
if [ $nb -gt $first ]
then
echo '#SBATCH --dependency=afterOK:'$jobid4' ' >> ${outputfolder}/Python_particle_n/Date${nb}/
              MeshToElmer_${nb}.sh
22     fi
23     echo '##SBATCH -p '$queue'
module purge
module load intel/13.1.0
module load intelmpi/4.1.0
module load mkl/11.0.2
module load hypre/2.9.0b
module load mumps/4.10.0
module load trilinos/11.0.3
module load elmer/latest
module load python-env
module load gmsh
python '$inputfolder'/Python_particle/MeshToElmer.py '$nb' '\'''$outputfolder''\''
' >> ${outputfolder}/Python_particle_n/Date${nb}/MeshToElmer_${nb}.sh
cd $outputfolder/Python_particle_n/Date$nb
echo sbatch ${outputfolder}'/Python_particle_n/Date'${nb}'/MeshToElmer_'${nb}.sh
jobid0=`sbatch  $outputfolder/Python_particle_n/Date$nb/MeshToElmer_$nb.sh | awk '{print $4}'`
echo "submitted '$jobid4' instance= '$nb'"
```

**Listing 1.** Excerpt from Bash shell script





scripts had been implemented as serial code and thus make use of one CPU core only, however, the submission code in `simu_coupling.sh` used same number of compute resources as for the other compute-intensive tasks, i.e. many CPU cores. Hence, while many CPU cores were reserved in the HPC cluster, only one CPU core was in fact used during these tasks while all the other reserved CPU cores were idle. The resource assignment was hence not optimal, in particular when

taking into account that these resource-wasting data conversion tasks occured in every iteration. Another example of inefficient resource usage is that the output files of one step – that were to be used immediately as input for the next step – were physically copied from the output folder of the current step to input folder of the next step, thus generating unnecessary I/O activity and consuming unnecessary storage space on the file system.

The initial workflow also required users to be acquainted with the target system tools and technologies: Firstly, they had

to understand the underlying batch system commands, i.e. how to craft the job submission scripts. Secondly, they had to manually trace the job status after the whole workflow execution came to an end. Thirdly, the `simu_coupling.sh` script was implemented using the Bash shell script language, whose knowledge was necessary to understand and enhance the application.

Finally, the script-based workflow enforced the sequencing of tasks through so-called job chaining (Line 21 in Listing 1), which is an intrinsic feature of the used batch system. However, there are two drawbacks when using this approach. Firstly,

while job chaining is provided by most HPC batch systems, the syntax and semantics vary from one batch job submission system to the other. Therefore, the implementation is tightly bound to the specific batch system environment and cannot be easily ported to other batch systems. Secondly, while all the jobs of the workflow, whether simple or iterative, can be easily controlled through job chaining when executing on one computing resource (e.g. one HPC cluster), this approach fails when more than one resource management system is involved in that workflow execution, e.g. by running the different steps on

different HPC clusters that reflect the different HPC characteristics of Elmer/Ice and HiDEM (with HiDEM benefiting from fast inter-connection needed for the MPI communication between the many CPU nodes, whereas the OpenMP implementation of Elmer/Ice benefits from many cores sharing memory on the same node).

## 5 Requirements Analysis

The problem analysis of the initial shell script-based workflow led to a set of requirements that aim at improving the workflow

with respect to usability, adaptability, maintainability, portability, robustness, resource usage, and overall runtime. Based on the weaknesses of the initial workflow implementation, we focused in particular on improving overall runtime, usability, portability and re-usability, and on enabling a uniform access in order to widen the scientific community that can use this glaciology workflow.

The requirements elicitation phase yielded the following requirements, which led to an improved design and implementation

of the workflow. (A summary of the requirements is provided in Table 1 together with a description of how each requirement is addressed and realised in our improved workflow.)

***R1: Readability and Understandability*** To continuously develop, maintain, and disseminate a scientific application for collaboration requires the implementation to have a clean, clearly modularized and error-free code. Since our case study





consists of many independent applications related to each workflow task, it is important that the identified tasks are well-segregated and do not contain overlapping tasks. A well-segregated workflow not only helps the application developer to further enhance the application, but also to distribute the code in order to collaborate with a larger scientific community.

**R2: Sequential Pipeline** The execution of jobs in the workflow should be orchestrated in a sequential manner such that one job step should not commence unless all previous steps are completed. Section 4 describes the side effects of the job chaining provided. This requirement envisages the whole scenario as a sequence of jobs that should connect all the involved applications in a batch-system-agnostic manner.

**R3: Dynamic Data Injection** The data injection for any workflow job should be transparent and easy to express. This requirement refers to the provisioning of data sets to individual workflow steps: Before a job is started, the required data needs to be available. Furthermore, dynamic data injection allows to import data from various sources using different protocols. A data-transfer-agnostic access is an add-on to this requirement.

**R4: Data Sharing Across Job Steps** The cost of data sharing across the jobs of the workflow steps becomes high when the data is replicated across each of the job steps. This unnecessarily consumes storage space and ultimately increases the overall workflow footprint in terms of resource consumption. Therefore, an adequate data sharing mechanism across the workflow should be available, which allows the simplified integration of data at application runtime; it will also facilitate optimal storage resource usage (e.g. of a parallel file system) in the target system. This is of particular importance when dealing with two different HPC clusters running different steps of the workflow, where data needs to be exchanged between the HPC clusters.

**R5: Resource-Agnostic Access** The continuum ice flow model (Elmer/Ice) is less resource-intensive than the calving model (HiDEM). This is due to very different spatial and temporal resolutions but also the models themselves, which require different amounts of computational resources (16 cores for the continuous ice flow model, but 560 cores for the discrete particle model). Getting access to CPU time on a small HPC cluster is typically easier than on big clusters, hence the workflow shall support the possibility to run these two significantly different steps on two different computing resources, thus reducing the amount of CPU and queueing time needed on the more powerful cluster. If the executions are running on heterogeneous clusters, there should be a layer of abstraction that encapsulates the intricacies of different resource management systems.

**R6: Parametric Execution** In our case study, most of the job steps need to be executed in an iterative way. Every new iteration takes input from a plain ASCII text file, called `n_list.txt`, that records the surface velocity data of some days of ice flow simulation. The requirement is to take the input data from the file and use it as a parameter for the iteration of the tasks. It means that the parameter value is shared by all the tasks involved. Furthermore, the iteration should not exceed a certain threshold.

**R7: Workflow Composition and Visual Editing** Many scientists are more comfortable with visual interfaces when linking many applications for one scenario, such as the presented glacier modelling workflow. It will be more robust for users to have a graphical interface that allows them to visually program and manage scientific workflows. In the glacier modelling scenario there are six main steps, each with different shell scripts and resource configurations, therefore a *Graphical User Interface* (GUI) can be very useful for visual editing, composition and automation of all the steps.





*R8: Workflow Tracing and Monitoring*  It should be possible to trace and monitor the whole workflow, including its sub-elements such as individual jobs. The extensive process of calving simulation may have to be aborted at some stage due to data or parameter anomalies. Therefore, it must be possible to interrupt the workflow at any point. Apart from that, real-time status of the jobs managed by the workflow should be provided.

5        *R9: Workflow Reproducibility* The workflow needs to support reproducing results by the original researchers and by third parties. If the created workflow is carefully designed and validated against any errors, it can be exported for re-use by a larger community. This includes not only exposing that workflow to a wider community on the same computational resource, but also running it in a completely different hardware or software environment (re-usability, adaptability, portability, and maintainability).

*R10: Secure Access* The workflow management system should be capable of providing an interface to let users run scientific workflows in a secure manner. This implies that adequate authentication and authorization need to be in place. This requirement further mandates the workflow system to be compatible with back-end computing clusters and existing production computing infrastructures.

*R11: Execution Environment Independence* This requirement supports a scenario which allows scientists to submit compu-
tations without knowledge of the parallel execution environment installed at the target computing resource. In our case, there are at least two different MPI execution environments involved, and thus two different MPI implementations. The intended middleware abstraction should not require a user to know the target environment that is providing the actual execution.

*R12: Data and Variable Configuration* Configuring required data elements such as workflow-centric input and output locations, and shared applications' environment variables or constants across many job steps can reduce much workflow man-
agement and development overhead. This may allow carrying out the design, execution, and debugging phases of many tasks in a more efficient manner. Therefore, in terms of overall usability and application maintenance, this requirement is considered important for realising the complex structure of connected tasks.

Any solution that addresses this set of requirements will make the scientific workflow usable for a wider set of communities working in glaciology.

## 6   Workflow Design

Using the initial shell script-based workflow as a starting point and taking the requirements R1–R12 into account, this section discusses the improvements and modifications to the initial workflow implementation.

Figure 4 shows the modified workflow composition. The steps are in fact very similar to the structure described in Section 3.3, and our design and implementation of an improved workflow is based on it. However, an important difference is that, initially,
there was one big shell script containing the whole workflow. Each of the steps shown in Fig. 4 has been extracted to obtain separate job scripts. In this manner, every step can separately specify different resource requirements and software capabilities. Furthermore, the number of lines of code in the scripts of each of the individual steps has been significantly reduced, thus addressing the general goals of improving adaptability and maintainability.



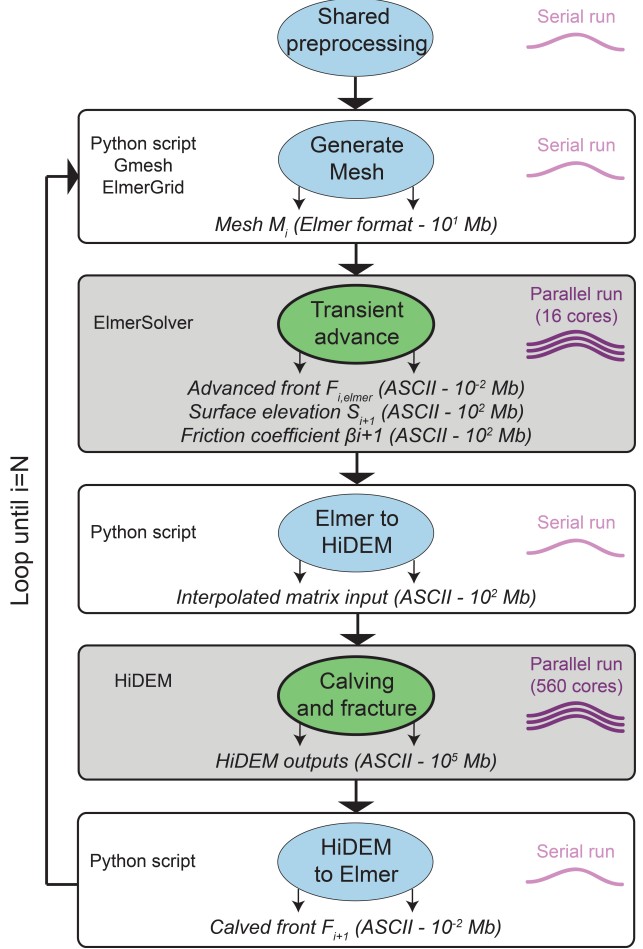

**Figure 4.** Whole workflow with blocks representing steps.

The data conversion tasks, such as "Elmer to HiDEM" and "HiDEM to Elmer" existed in the initial workflow implementation as part of the respective ElmerSolver and HiDEM jobs. As the latter are both resource intensive (i.e., they run on multiple cores), but the data conversion tasks are serial and require less resources, it is inappropriate to reserve (and thus waste) parallel resources for the serial data conversion. The separation of tasks in our improved workflow design enables us to configure them

5   to use only a single core, as sufficient for their serial execution.

The step "shared preprocessing" is introduced as an additional task to manage the initialisation phase of the workflow. It mainly provides the involved applications with the required initial input data sets and prepares shared output directories where the subsequent individual workflow steps accumulate intermediate and final results. In this step, the shared workflow variables are also initialized, and the required intermediate working directories are created. Without input and output storage

10   management and shared variables, our workflow implementation would stall. Therefore, this step is a necessary prerequisite for all further iterations.





## 7 Workflow Implementation

While the re-designed workflow could now be implemented again as a shell script, doing so would ignore most requirements described in Section 5. Hence, we use the *Uniform Interface to Computing Resources* (UNICORE) middleware (which includes a workflow engine) to automate and implement our workflow. We have contributed to the creation of both UNICORE in general (Memon et al., 2007) and the workflow engine in particular (Memon et al., 2013b).

### 7.1 UNICORE Foundations

UNICORE (Streit et al., 2010) is a distributed computing middleware that provides abstractions for job submission and management on different kinds of job scheduling systems. Hence, jobs can be submitted to a cluster without needing to know about the job scheduling system internally used by that cluster. The abstraction is achieved through a unified set of interfaces that enable scientists to submit computation jobs without considering any intricacies of the underlying batch system. UNICORE takes care of automatic translation of job requests to multiple target resource environments.

UNICORE provides a workflow system based on the *Service Oriented Architecture* (SOA), i.e., all the main functional interfaces of the workflow system are exposed as web services. Figure 5 gives a holistic view of UNICORE's multi-layered architecture that is composed of Client, Server, and Target System tiers. The Client tier has two main variants, the *UNICORE Command-line Client* (UCC) and *UNICORE Rich Client* (URC). However, other third party client applications such as scientific gateways, science portals and client APIs can also be integrated, if they comply with the provided server-side interfaces.

To address the goal of usability, we put emphasis on the URC, which is an Eclipse-based (Eclipse Foundation, 2013) client application implemented in Java. It provides users with a wide range of functionalities such as workflow management and monitoring, data down- and upload to a remote cluster, a GUI for workflow editing and resource and environment selection panels. More details are provided later in Section 7.1.1.

On the server side, UNICORE has a set of web service interfaces, which are based on the *Web Service Resource Framework* (Graham et al., 2006) and WS-Addressing (Box et al., 2004) standards. The server side is further divided into two layers, the *workflow system* and *atomic services*. The workflow system provides interfaces to manage, orchestrate, broker, and debug workflows. Scientific workflows can be composed of multiple jobs modelled as graphs. The atomic services layer allows handling of individual jobs, which includes their execution, data transfer, authorization and user access. The main web services responsible for executing and enacting scientific workflows are the *Workflow Engine* and *Service Orchestrator*.

Before a web service request is forwarded to the real resource environment, it has to pass through the *eXtended Network Job Supervisor* (XNJS) and *Target System Interface* (TSI) components. The XNJS is a robust execution engine that is responsible for forwarding job submission requests to the target resource management system through the *Target System Interface* (TSI). The TSI is a low-memory-footprint component and is deployed on the cluster's login node. It performs the actual translation of the XNJS request into the job command script format used on the particular system. As every batch system may have a different job command language, the TSI is very helpful in encapsulating the batch system it interacts with. For this purpose, the TSI supports most production batch systems, such as SLURM, Torque, and Load Leveler.



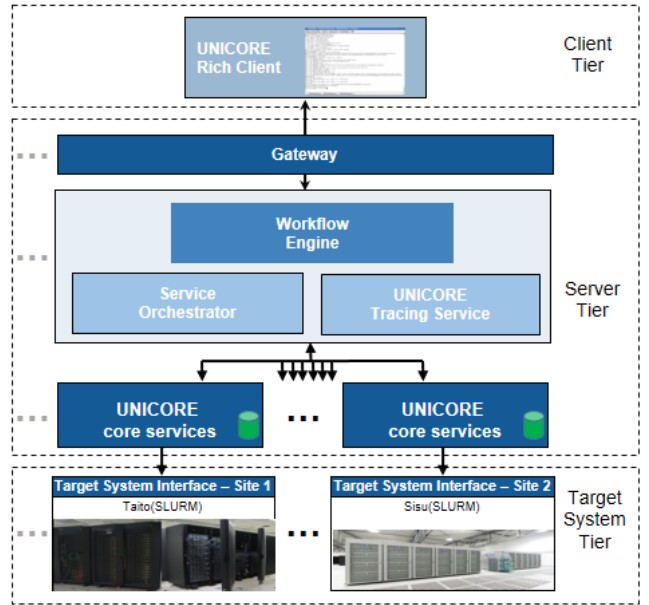

**Figure 5.** Multi-tiered UNICORE architecture.

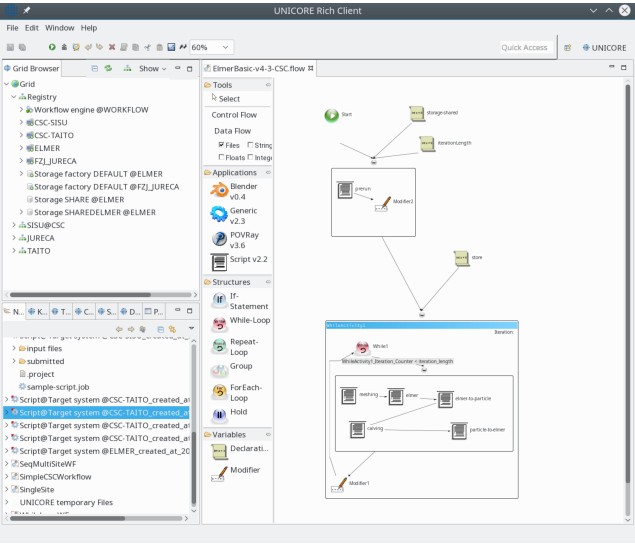

**Figure 6.** The UNICORE Rich Client workbench.

### 7.1.1 *UNICORE Rich Client*

The *UNICORE Rich Client* (URC) provides a rich GUI to access and interact with remote compute resources, jobs, and data (Demuth et al., 2010). It is based on Eclipse RCP, the *Rich Client Platform* (Eclipse Foundation, 2013) and communi-



cates to the UNICORE server-side components and services (Workflow management, atomic services, etc.) through a secure communication channel. The URC helps end users who do not want to know the system specifics and want to graphically create workflows, which is the recommended way for crafting complex flows of tasks. As the UNICORE middleware provides HPC-oriented functionalities, it would be a daunting task to expose them to scientists with minimal Linux or HPC background.

Considering that, the URC exposes most of the middleware functionality in a user-friendly manner.

An example screenshot of the URC workbench is shown in Fig. 6. The main features provided by the URC are:

**Grid browsing:** Viewing and interacting with all services in a hierarchical manner.

**Workflow editing:** A fully-featured editing canvas, on which remote job applications and powerful constructs like loops (FOR-EACH) and conditional statements (IF-THEN-ELSE) can be visually composed.

**Security configuration:** Management of multiple X.509 credentials and trusted certificates at one location.

**Workflow monitoring and tracing:** When a single job or a complex workflow has been submitted, a separate graphical layout appears that shows the real-time status of the execution. In addition, the resource status is automatically updated on the grid browsing panel while the job proceeds towards its final state.

### 7.2 Workflow Realisation using UNICORE

The shell-script-based initial workflow described in Section 3.3 uses no middleware abstraction, but has low-level hard-coded dependencies on the assumed resource environment. This old implementation violates several of our requirements related to system interaction, job submission and control, and data management. I also exhibits several usability issues. Considering the complexity of the compute and data aspects, modifying the initial workflow management script to satisfy our requirements R1–R12 would take tremendous effort if no abstractions and high-level concepts (such as those provided by UNICORE) were used.

We therefore employ the UNICORE workflow management system to automate the workflow of our use case in a high-level way.

To improve usability, the new, improved workflow was designed using the visual editor provided by the URC. The editor allows scientists to visually drag and drop different task types for different application types that may be enclosed in conditional structures. The supported task types are simplified, and small Bash shell scripts containing customized or generic applications

can be executed remotely on user-specified resources.

Figure 7 shows the sequence of the workflow tasks defined for our glacier modelling case study. The major steps described in Section 3.3 can be directly mapped to the tasks defined at the URC level. However, there is an additional step required for arranging any intermediate input and output data sets. The concrete tasks defined at the URC tier are:

1. prerun: This initial task declares global constants for all the workflow instances and also creates a central output directory
that is shared across the jobs participating in an instance of the workflow. It also sets an initial input that contains the total number of iterations. The initial input data takes the observations of one year with an interval of 11 days.





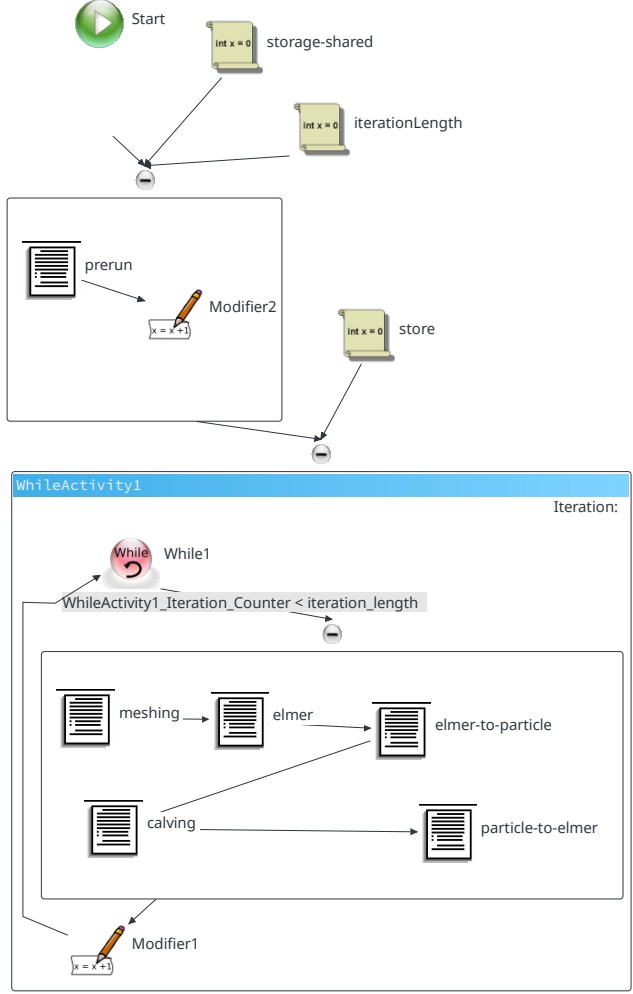

**Figure 7.** Graphical workflow snippet from the URC workbench.

2. meshing: This task generates a 2D mesh by running the Gmsh and ElmerGrid tools sequentially using a set of input files. ElmerGrid is a mesh generator that takes as input a mesh from Gmsh to generate the output mesh to be used in the following step by Elmer/Ice.

3. elmer: In this task, the finite element modelling is executed. In order to achieve that, the parallel `ElmerSolver` application is invoked. This application is based on MPI/OpenMPI standards and thus typically runs in HPC environments.

4. elmer-to-particle: This is a Python script that transforms the Elmer/Ice output produced by the previous step to create a format required for the HiDEM application.





5. calving: This task performs discrete element modelling by running the HiDEM application. This step consumes much more resources than any other task of the workflow, therefore the converted data has to be carefully injected to this phase that uses MPI to run in huge HPC environments.

6. particle-to-elmer: In this task, the output of the HiDEM job is transformed by a Python script into the format used by task 2, i.e. (re-)meshing for Elmer/Ice.

In addition to the task definitions, a shared variable is required that contains the workflow output location that is to be used across all the tasks. This is the only variable meant to be customized for a different user: In case another user wants to run the same workflow on the same set of resources, e.g. the same HPC cluster, then this single value has to be adjusted to the preferred file storage location.

Prior to the workflow execution by URC, the user has to: 1) configure on which target site each task runs; 2) specify the extent of computing resources it requires; 3) provide a list of input and output files involved. Once the tasks are prepared, the workflow can be submitted for execution on (remote) HPC clusters. During the workflow execution phase, the sequence of running tasks will follow the workflow graph specified by the user.

Listing 2 shows three snippets. First, the elmer and particle-to-elmer task execution: For both tasks, it can be seen that only executables are invoked without including batch-system-specific statements (lines 4 and 9) as was the case in the initial shell script (cf. Listing 1). The URC version in fact contains only the workflow- and application-specific elements. Finally, Listing 2 also shows the common code section (lines 13–17) that fetches the value of the last iteration required to process the data of the current iteration. This common code snippet must be present in all the workflow steps except the prerun step. In the given scenario, these lines have to be replicated in every step enclosed in the iteration block. Practically, this is just syntactic overhead, and considered negligible when the number of affected steps is small.

## 7.3 Resource Setup and Interaction Scenario

Our new and improved workflow requires the deployment of separate UNICORE server and client instances. In order to compare our new workflow implementation with the initial shell-script-based workflow implementation, the same back-end resources are used: The server-side deployment spans two production clusters at CSC in Finland, namely the Taito cluster for smaller jobs such as Elmer/Ice and the bigger Sisu cluster for massively parallel jobs such as HiDEM. On both sites, the UNICORE instances were separately deployed. These sites already have SLURM as a resource management system available, but with different hardware configurations: Taito has heterogeneous node groups with varying capabilities and CPU layouts (Haswell and Sandybridge), whereas Sisu has a symmetric configuration with all the nodes providing same number of processing, data and memory resources. Furthermore, as some sort of master, a shared UNICORE workflow management and a resource broker instance have been deployed on a cloud computing instance running at the Jülich Supercomputing Centre (JSC) in Germany.

The resources at CSC needed to have the respective applications installed to support the complete workflow execution, i.e. on Taito, the Elmer suite with the Elmer/Ice glaciology extension was installed, whereas the particle calving application (HiDEM)



```
elmer:
# creating task-specific folders
echo $geometryname$nb.sif > ELMERSOLVER_STARTINFO
srun $ELMERBIN
# accumulate output and move to the main workflow output
particle-to-elmer:
# creating task-specific folders
python $JOBWD/input/Python_particle/ParticleToElmer.py $nb $JOBWD
# accumulate output and move to the main workflow output
common code:
if [ $currentctr -gt 1 ]; then
nrbefore=`sed -n -e "$((COUNTER))p" $JOBWD/input/n_list.bak`
nb_before=`awk '{print $1}' <<< $nrbefore`
firstIteration=false
fi
```

**Listing 2.** Excerpt of the elmer and particle-to-elmer tasks, and the common script snippets implemented in URC

was provided on Sisu. In addition to these executables, a Python environment had to be available on both systems for running the Elmer-to-Particle (and vice versa) conversion scripts.

Figure 8 depicts the deployment perspective of the site configurations. While the steps (1) to (7), marked in the figure as circles, will be explained later, the general deployment prerequisites are as follows:

The workflow development and remote execution is managed by the user through the *UNICORE Rich Client* (URC). It is thus required to have the URC installed on the user side. To obtain access to the remote UNICORE sites at CSC, the user has to acquire X.509 credentials and trusted certificates of the server instances. By using these credentials, the user can interact with all the UNICORE server instances of the infrastructure that she has access to.

After the credentials are set up, the user has to add sites that shall be used for the workflow execution. For this, only a single
location of the discovery service called Registry is provided. If the user's identity is known to the Registry instance and all the concerned compute sites, then these sites will be available for execution. Following the discovery service inclusion, the workflow development and management can be performed easily.

Figure 8 depicts all the architectural elements and the workflow submission steps that are taking part in the glacier modelling use case. The top layer is the client tier, i.e. the URC operated by a user who creates and runs the workflow instance. In Step (1),
the user discovers any available Workflow Engine instance, and selects which instance will take care of the whole workflow execution. The requests to the Workflow Engine instance are encoded in XML, with the individual job requests of a workflow being compliant to the JSDL specifications (Anjomshoaa et al., 2008; Savva, 2007). When the user request is received by a Workflow Engine instance (Step (2)), it validates the incoming message and checks whether active sites are available, i.e.



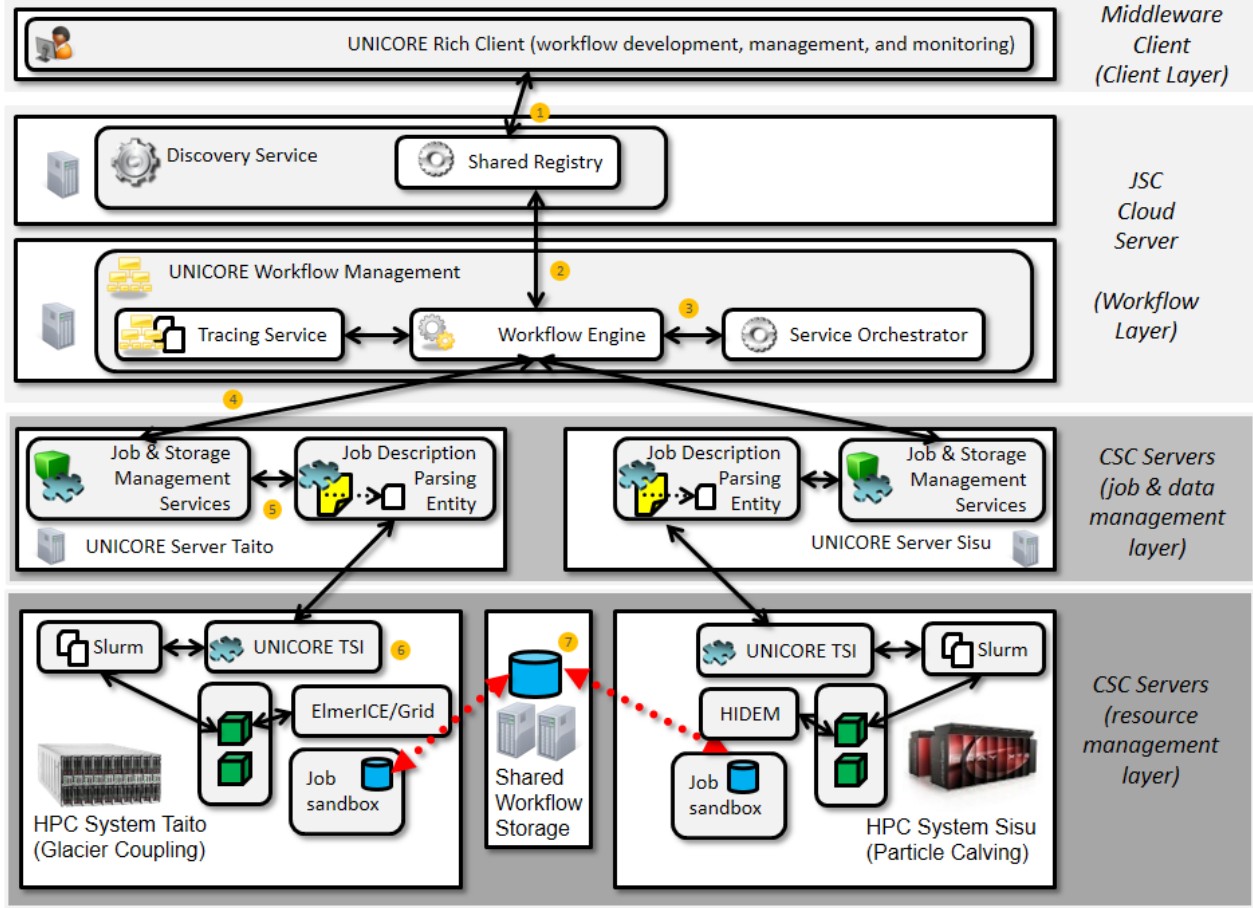

**Figure 8.** Deployment of UNICORE client, workflow and job management services on multiple sites.

whether UNICORE server instances are ready to run and manage single atomic jobs that are part of the workflow request. After the successful validation of the request, the Workflow Engine contacts the Service Orchestrator in Step (3), which acts as a resource broker for many UNICORE servers. The Service Orchestrator submits individual job requests to the sites matching with each job's individual resource requirements (Step (4)). The UNICORE server at each of the given sites (Sisu and Taito

5    in our case) receives the request from the Workflow Engine instance (Step (5)), and dispatches it further to the remote batch system environment. Both clusters maintain a separate TSI daemon service that translates in Step (6) the request coming from the UNICORE server to the batch-system- and environment-specific job command scripts. The data used by the glacier modelling use case is shared and re-used through the central workflow storage management service provided by the UNICORE server (Step (7)).

10    Figure 9 shows the output of the calving task that is the last step of each workflow iteration. Without the UNICORE-based workflow automation, it was difficult to manage the overall makespan of this step. After transforming it using the UNICORE



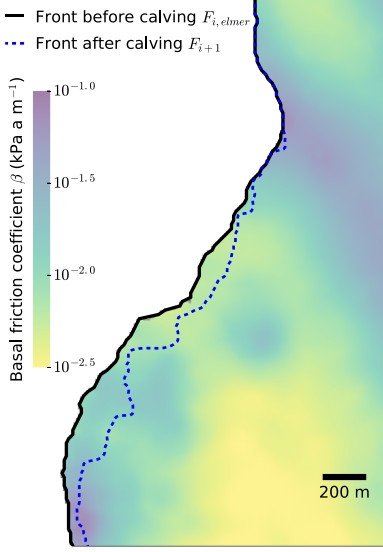

**Figure 9.** Basal friction coefficient, $\beta$ and new position of the front $F_{i+1}$ after calving from the HiDEM simulation generated through the UNICORE-based implementation.

system, the management became easier and seamless as the calving tasks are only invoked when the preceding phases (such as coupling and data conversion) and iterations were completed successfully, i.e. when they could provide reasonable input to the computationally expensive HiDEM executable.

## 8  Discussion

This section discusses the most vital elements of the glacier coupling and calving case study to show that the main goal was achieved, i.e. to provide users with simplified access to distributed HPC resources.

In the implementation, the application environment and the dataset used were the same as before, with the addition of the UNICORE layer. Therefore, this discussion does not cover the application performance and scalability, but rather the overall workflow effectiveness, robustness, and usability.

### 8.1  Fulfilment of Requirements

In this section, we show how the requirements presented in Section 5 have been fulfilled through the use of the UNICORE workflow management system. Table 1 lists each of the requirements and explains briefly how it has been realised. The details are as follows:

Requirements *R1 – Readability and usability* and *R7 – Workflow composition and visual editing* are addressed because the URC client comes with a rich workflow editor that allows simplified access to multiple resources and helps users to compose and connect workflow tasks in an easy manner.





**Table 1.** Summary of requirements and how the new workflow implementation addresses them.

|     | *Description* | *Realisation* |
| --- | --- | --- |
| *R1* | Readability and usability | URC workflow editor |
| *R2* | Sequential pipeline | Workflow management and enactment |
| *R3* | Dynamic data injection | Automatic data import and export |
| *R4* | Data sharing across job steps | UNICORE's data management services |
| *R5* | Resource-agnostic access | UNICORE's job management services |
| *R6* | Parametric execution | Composite constructs and loops |
| *R7* | Workflow composition and visual editing | URC workflow editor and widgets |
| *R8* | Workflow tracing and monitoring | UNICORE's workflow tracing and management services |
| *R9* | Workflow reproducibility | URC's project export wizard |
| *R10* | Secure access | PKI, X.509, and mutual authentication |
| *R11* | Execution environment independence | Job incarnation through XNJS and TSI |
| *R12* | Data and variable configuration | Middleware-supported variable resolution |

The UNICORE WMS provides sequential access by enforcing a barrier on a workflow task that is about to be processed until its preceding task completes successfully. This supports requirement *R2 – Sequential pipeline*.

The workflow's data management is considered to be an essential requirement for any data intensive application. It is typically either a remote data transfer or data movement within the file system used by that computational resource. The workflow management system should not bother the user with this. In our case, the UNICORE atomic services take care of any data movement to a third party data space or a local cluster; the user is only expected to specify the source and target file locations. After the workflow has been submitted, the required data transfers are carried out automatically. This functionality supports requirements *R3 – Dynamic data injection* and *R4 – Data sharing across multiple job steps*.

For the glacier modelling workflow, the UNICORE-based implementation executes Steps 2-6 of the workflow in a WHILE loop until a certain number of observations has been reached. As the observations are stored in a file, they need to be processed and the values need to be loaded to UNICORE's WHILE loop variable store. Each time the workflow instance is created and submitted, the loop construct loads the file called `n_list.txt` and takes each observation from that file to run the underlying steps in a parametric way. This feature supports requirement *R6 – Parametric execution*.



The UNICORE-based glacier coupling and calving workflow is using the computing resources deployed on CSC's Sisu and Taito clusters. If a future user of this application intends to deploy and run the workflow in a different resource and application environment, this will be possible with minimal effort: The UNICORE atomic services provide a layer of abstraction over multiple batch systems, which fulfils requirements *R5 – Resource agnostic access* and *R11 – Execution environment independence*.

In case another user is interested in using the UNICORE-based implementation, URC provides a feature to export the workflow in a reproducible format to be re-usable by other users. This supports requirement *R9 – Workflow reproducibility* (more details on workflow reproducibility are discussed later in this section).

The URC interface allows users to specify any application-, data- and environment-specific variables, scoped either to one task or a group of tasks. To enhance and simplify our new workflow implementation, a number of workflow-wide and application-specific variables were used. During workflow runtime, they are resolved without needing any user intervention. This addresses requirement *R12 – Data and variable configuration*.

Requirement *R10 – Secure access* is essential since the workflow will have access to large and precious compute resources, for which the UNICORE-based deployment ensures secure interaction between the user and all the services she communicates with, such as workflow executions and access to storage and data. Users accessing any remote UNICORE-based services are required to possess X.509 credentials in order to use these services.

Finally, to compose, manage and monitor workflow submissions interactively, URC provides a separate visual interface to edit or create individual workflow tasks or monitor running workflows and the involved jobs. This supports requirement *R8 – Workflow tracing and monitoring*.

## 8.2 Middleware Deployment Overhead

For the UNICORE-based implementations of computational applications, the provider of the computational resources has to ensure the availability of UNICORE server-side and URC deployments. Maintaining a server-side deployment is not trivial because it needs a dedicated server that manages workflows and atomic jobs. Even though the URC is a rich client with a GUI, it is Java-based and thus platform-independent, and does not have any installation overhead. However, every URC user needs to have a credential store based on a Java keystore and personal X.509 certificates, which may be confusing for new users. On the other hand, a shell-script-based implementation would require SSH-based remote access, which may cause inconvenience for some users who are not used to interact with systems through a command-line interface.

## 8.3 Modularization

The initial shell-script-based workflow implementation consisted of a single, huge Bash shell script, including call-outs to all the related tasks (whether serial or parallel), defined in a monolithic way without proper separation. Hence, one tiny change may have caused unexpected side effects resulting in errors. For every type of task, the resource requirements (e.g. number of CPU cores) were also maintained there. The monitoring of the tasks was very cumbersome due to the Bash script execution that involves nested scripts.




On the other hand, the UNICORE-based implementation using URC allows us to cleanly separate the tasks in a modular way, which enables us to individually monitor and manage tasks even while they are in the execution phase. The complete workflow management can be performed interactively and visually through the URC's GUI. Our experience is that using the URC is less error prone than the purely shell-script-based approach.

## 8.4 Workflow Extensions

In practice, enhancements or addition of new scientific methods to an existing workflow are inevitable. Our experience with the shell-script-based workflow was that even minor changes are tedious, errors may occur easily, and try-and-error debugging wastes CPU hours on HPC resources. In contrast, our experience with the UNICORE-based workflow implementation shows that it is easy to edit the workflow, and validation of individual tasks is possible even before the whole workflow is executed on HPC resources.

For illustration, let us assume that we want to introduce some additional tasks, e.g. on a different HPC resource than the one already used. For the shell-script-based approach, this would require a drastic change of the main script, whereas in the UNICORE-based approach, it just requires the definition of yet another step (such as a script task) using drag-and-drop, followed by resource configuration using a GUI dialogue. Both are easy to achieve using the URC.

## 8.5 Resource Abstraction

In the shell-script-based workflow implementation, the job submission command scripts for the concerned batch systems were hard-coded and distributed over many, scattered locations of the shell script. UNICORE-based job submission does not require such a hard-coded approach, as job submission statements are automatically created on the server side by UNICORE, on behalf of the user. However, while this middleware abstraction offers user-friendliness, it also means that, due to the abstraction, some batch-system-specific low-level technical features which might be useful for the application cannot be fully utilized, as the middleware layer hides them in the pursuit of unified and consistent experience. In our case study, we needed only standard job submission features, so this was not a problem.

## 8.6 Manual WHILE Loop Configuration

One issue with the URC version used for our workflow was that it was not possible to create the WHILE loop configuration needed for the iterative coupling using the URC GUI – instead, textual scripting was required as a workaround. However, the latest URC version already supports in an easy manner the automatic loading and processing of text files to be used as properties of the WHILE loop provided by the UNICORE workflow engine. Future versions of our workflow implementation will adopt this feature offered by the latest URC release.



## 8.7 Reproducibility and Re-use

As in every field of science, it is necessary that other scientists are able to reproduce and verify the glaciology results we obtained. It is however not trivial to ensure this in a different resource environment, e.g. using a different HPC cluster or batch system.

In the shell-script-based workflow scenario, the Python scripts responsible for running the conversion routines, such as, Elmer-to-Particle or Particle-to-Elmer, were using system dependent paths to the applications used for translating data during every iteration. The conversion steps will lead to failure if the file system or path to the applications changes. In the UNICORE-based implementation (where the batch system translation is automated as described in Section 8.5), this scenario is much simplified for the end users, and for them just the UNICORE workflow needs to be exported into a re-usable workflow (using

an XML format). The exported version can easily be imported by any other UNICORE system. The only requirement is that the imported workflow tasks' resource requirements and the shared workflow variables have to be re-aligned to the new environment (that might, e.g., have a lower number of CPU cores). If the cluster environment stays the same, but only the user changes, there is even no need to re-configure target resource requirements, but only shared workflow variables concerning the storage of user-specific data, e.g. the location of the datasets to be used as input needs to be adjusted. In addition to

reproducing experiments, the high-level UNICORE workflows can also be easily extended (cf. Section 8.4) by other scientists from the glaciology community to adapt them to their needs.

## 9 Related Work

There is a wide variety of work and research on scientific workflows that couple different computing-intensive applications in many disciplines. However, new research fields often pop up, recently e.g. data-intensive applications such as machine

learning, data mining, or statistical analysis. As a consequence, the WMS technologies are often evolved to fit the needs of specific user communities or hardware resources that in turn makes them too distinct to share across systems or a broad set of user communities (Ferreira da Silva et al., 2017).

    One known example is Taverna (Wolstencroft et al., 2013), which in principle is a general WMS, but is significantly driven by bio-informatics communities with the need for *High-Throughput Computing* (HTC)-driven "-omics" analyses (proteomics,

transcriptomics, etc.) and thus lacks distinct support of cutting-edge HPC systems such as those used in our case study. At the same time one can observe that HPC systems quickly evolve with an ever new set of features that need to be adopted to ensure good performance of the applications, thus motivating the use of the UNICORE WMS that is specifically designed for HPC environments and constantly maintained and tuned to new HPC system designs.

    Having a closer look in the Earth science domain, another field of related work are scientific gateways such as the Southern

California Earthquake Center (SCEC) Earthworks portal that inherently adopts the WMS system Pegasus (Deelman et al., 2015) and DAGMan (Frey, 2003) in order to run on systems provided by the US infrastructure Extreme Science and Engineering Discovery Environment (XSEDE). Pegasus itself is just one component on top of DAGMan that in turn is based on the



HTCondor middleware for HTC, which in our review did not meet the full capabilities required for HPC in general and the large supercomputers used in our study in particular.

Given the nature of the scientific gateway of the Earthworks portal and its focus on specifically only supporting earthquake science applications, it is not possible to re-use the system for our research in glaciological models and the above-described coupling approaches using European HPC systems.

Reviewing related work further leads to consideration of older WMS systems such as VisTrails (Callahan et al., 2006). Work has been done there in the past to work with emerging HPC systems, but maintenance is typically lacking today. There are also adaptations of VisTrails to specific environments such as the NASA Earth Exchange (NEX) collaboration platform, as described by Zhang et al. (2013).

To the best of our knowledge, there is no general WMS that is both mostly used with distinct workflow features (as outlined above) in the Earth science domain, and is also applicable to cutting-edge HPC systems, as required in our study to perform the efficient calculations.

## 10  Conclusions

Scientific workflows automate and enable complex scientific computational scenarios, which include data intensive scenarios, parametric executions, and interactive simulations. In this article, a glacier ice flow and calving model have been combined into a single high-level scientific workflow. The ice flow is solved using the *Finite Element Model* (FEM) code Elmer, with glaciological extensions called Elmer/Ice, whereas calving was simulated by the *Helsinki Discrete Element Model* (HiDEM). While the underlying initial workflow had already been automated, that had been done via a large shell script that was hard to understand, use, debug, change, port, maintain, and monitor. Furthermore, it was not robust with respect to runtime failures that can occur at many places of an HPC system.

Hence, we created a new workflow implementation based on the state-of-the-art UNICORE middleware suite. The workflow can easily be composed on a high and abstract level through the *UNICORE Rich Client* (URC)'s visual workflow editor. The workflow developed this way can be submitted in a fire-and-forget pattern to HPC clusters, while real-time monitoring is also possible. For this case study, the production deployment of UNICORE instances on CSC's clusters Taito and Sisu has been used.

We evaluated our new workflow implementation from different points of view, such as workflow extensions, usability, and variable resource access. Most notably, the workflow implementation can be exported to an abstract machine-readable format, so that other users interested in reproducing results can easily re-run the simulation on a different platform and still obtain the same outputs generated by our experiment, or they can re-use the workflow to adapt and apply it to new datasets. Furthermore, this UNICORE-based workflow can be easily changed by simply updating the existing tasks, given they are properly configured with respect to the needed target resources. The workflow can also be easily extended due to the inherent high-level of abstraction.



While we demonstrated the workflow management system approach based on the glacier modelling case study, we believe that the requirements that we derived from our use case are in fact applicable to many other Earth Science modelling applications, and even to further scientific disciplines that involve distributed HPC resources. Hence, our UNICORE-based approach appears promising also for other e-Science cases.

With regard to future work, it would be possible to create a portal that provides even simpler access to our glaciology workflow. This might include a web-based interface where a scientist simply has to upload her credentials for using an HPC cluster, upload or point to the dataset to be used, configure some simulation-specific parameters, and finally press a button to start the workflow on one or multiple HPC clusters.

As the UNICORE workflow management system is independent from any scientific domain, we encourage other disci-
plines to transform and automate their e-Science steps into automated workflows. We have successfully demonstrated this already for applications in remote sensing (Memon et al., 2018) and for interpretation of analytical ultracentrifugation experiments (Memon et al., 2013a).

*Code availability.*  Elmer and Elmer/ICE sources are freely available at https://github.com/ElmerCSC/elmerfem. The initial coupling (shell) scripts can be obtained by directly contacting the author. HiDEM sources can be openly accessed via https://doi.org/10.5281/zenodo.1252379.
All the UNICORE packages are available at http://unicore.eu. The UNICORE-based glacier model coupling workflow that can be used as template and associated Bash helper scripts are available via http://doi.org/10.23728/b2share.f10fd88bcce240fb9c8c4149c130a0d5.

To access and run the workflow, the UNICORE sites and the workflow services, and the application packages and the coupling scripts have to be installed. The intended users need to have appropriate and valid X.509 credentials, because *UNICORE Rich Client* (URC) requires them to present the credentials while accessing the workflow services.

*Author contributions.*  SM has developed the overall workflow design, prototyping, development and realisation, which also includes the workflow deployment and administration. He has also streamlined the glacier modelling use case scripts and application deployment on CSC clusters. DV contributed with the scientific problem that motivated the construction of this workflow. She has also developed the coupling script, and provided significant contributions during the workflow development, testing and deployment phases. TZ is heavily involved in the Elmer/Ice project and provided his support in the integration and deployment of Elmer/Ice. JÅ is the main author of HiDEM. HN provided
input on workflow foundations, conclusions, and also worked on streamlining the manuscript. In addition, he was in charge of the application to NeIC Dellingr for CPU time on CSC clusters. MR has provided related work in highlighting scientific workflows and Earth science modelling. MB has contributed to the system architecture from the software engineering perspective.

*Competing interests.*  The authors declare that they have no conflict of interest.

*Acknowledgements.*  This work was supported by NordForsk as part of the Nordic Center of Excellence (NCoE) eSTICC (eScience Tools for
Investigating Climate Change at High Northern Latitudes) and the NCoE SVALI (Stability and Variation of Arctic Land Ice). The authors wish to acknowledge CSC – IT Center for Science, Finland, for computational resources that were granted via the Nordic e-Infrastructure Collaboration (NeIC) Dellingr project. The authors are grateful to Jaakko Leinonen (CSC – IT Center for Scinece) for his support and patience in making the UNICORE services available on CSC's computing resources and to Joe Todd (University of St. Andrews) for publishing HiDEM on GitHub.



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
