# Peer review of "Scientific Workflows Applied to the Coupling of a Continuum (Elmer v8.3) and a Discrete Element (HiDEM v1.0) Ice Dynamic Model"

_Geoscientific Model Development, 2018_

## Referee Comment (RC1) · Anonymous Referee #1 · 16 Aug 2018

In their manuscript, Memon et al. describe the use of the workflow management system (WMS) UNICORE for modularizing the coupling between the ice flow model Elmer and the Calving model HiDEM. According to the manuscript, this coupling has previously been done with one 400-line long shell script calling various processing scripts and the models. The paper now develops requirements for a rewrite of this coupling that are met by the WMS UNICORE, that Memon propagates in several of his previous publications. Some of these requirements seem constructed to match the specifications of the WMS, and not arising from the reality of modeling (see below). I am not convinced that the 27-page manuscript about the conversion of a 400-line (∼10 pages in this format) shell-script into a high-level system, is advance in Geoscientific Model

[Figure]

Development, and suggest rejection of the manuscript.

Having coupled models - even across different supercomputers - before, I can definitely see a great value in easing this process. UNICORE may be a contribution to solving difficulties arising here, and with the portability of solutions, although, telling from the manuscript, the installation of UNICORE on a supercomputer seems to be far from trivial ("Maintaining a server-side deployment is not trivial because it needs a dedicated server that manages workflows and atomic jobs" (p. 23 ll 22), "The authors are grateful to [. . .] for his support and patience in making the UNICORE services available of CSC's computing resources"). Furthermore, my personal experience tells that getting the models themselves to run on different supercomputers tends to be the agonizing part, with scripts being no match, however poorly they may be written. In the end, the authors have managed to replace 400 lines of shell script (including job-headers) with a dependency on a high-level workflow management system with some services hosted at yet another supercomputing center. I am not convinced that a thorough clean-up/rewrite of the shell script would not have solved the main problems of the old script with way less effort and overhead.

The manuscript itself is somehow entangled between the scientific aspects of the ice dynamical problem, and the technical aspects of the software solution for job control on a set of supercomputers. It lacks a clear focus on either side, as can easily be seen from the mixture of plots that describe the glaciological problem (Figs. 1, 2, 3, 9) and the software solution for abstracting the scripting (Figs. 4-8, with Fig. 7 duplicating the right half of Fig 6). While we are introduced to the basic reasons why one would run the different glaciological models for the different sub-problems, and get some insights on the model grids, and which piece of code solves which sub-problem, there is no interpretation of the results shown in fig 8, or direct glaciological relevance of re-writing the coupling. On the other hand, for the description of how the authors took a shell script and turned it into something more high-level, the details of the glaciological problem are largely irrelevant.

Some of the complaints against the shell script solution are rather surprising to read. On page 10 lines 11f the authors complain "Thirdly, the simu_coupling.sh script was implemented using the Bash shell script language, whose knowledge was necessary to understand and enhance the application." Similarly, on page 23 lines 25-27 "On the other hand, a shell-script-based implementation would require SSH-based remote access, which may cause inconvenience for some users who are not used to interact with systems through a command-line interface." I find it hard to image that somebody would successfully use ELMER on a supercomputer without being able to read and manipulate a shell script. The fact that all sub-steps in their new solution are done in shell or python scripts, is ignored. Similarly I don't see a problem with the user having to take care of the code that copies files from one supercomputer to the other, or to adjust a job header for a new computer/queuing system. These tasks are usually adjusted within a day, with individual tasks being on the order of 10 minutes for anybody used to the supercomputer.

Reading the manuscript, the reader is taken through the same processes that the authors must have gone through when re-working the coupling script(s). The presentation of the initial state as well as the solution duplicates a lot of the content of the manuscript, with the second presentation being better structured than the first one, just as the scripts have gained in structure. The detailed specifications of the I/O and jobs of all sub-components are important for the author of this script, but rather belong into a manual than into a scientific manuscript.

In the end, the authors present a solution with different model parts being run on different supercomputers so each of them can run with maximum efficiency. Sadly, we never learn, how much resources each of these bits requires and whether there is a point in increasing the complexity of the problem to two supercomputers with remote file transfers for gaining computational efficiency, or whether simply letting the smaller model run in a separate (slightly inefficient) job on the same computer would have been faster in the end. While the authors claim a focus of their work was on improving

performance, (p.10 l26 "focused in particular on improving overall runtime [,...]"), they never provide any information about the results of this endeavor in terms of reduction of overall runtime, or similar metrics.

---

## Author Comment (AC1) · 27 Aug 2018

**Reply to reviewer 1 on "Interactive comment on "Scientific Workflows Applied to the Coupling of a Continuum (Elmer v8.3) and a Discrete Element (HiDEM v1.0) Ice Dynamic Model"**

We are very grateful for the reviewer's comments towards improving our manuscript, and would like to respond to the main points below:
* * *
Reviewer:

"UNICORE may be a contribution to solving difficulties arising here, and with the portability of solutions, although, telling from the manuscript, the installation of UNICORE on a supercomputer seems to be far from trivial ("Maintaining a server-side deployment is not trivial because it needs a dedicated server that manages workflows and atomic jobs" (p. 23 ll 22), "The authors are grateful to [. . .] for his support and patience in making the UNICORE services available of CSC's computing resources").

Reply:

We would like to disagree on this point. While the UNICORE workflow deployment certainly is a prerequisite to our development, installation and configuration of any middleware - just like the time-intensive maintenance of HPC systems in general - is not the task of the user, but rather that of administrators at HPC centres (to whom we expressed our gratitude). The most challenging part was to study and analyse the coupling scenario as a whole, and then segregate the tasks from the available set of scripts and group them in different workflow elements so that any addition of new tasks in the future should not hurt the existing structure. UNICORE middleware in our application acts as an abstraction, i.e. a general-purpose HPC middleware solution to abstract multiple kinds of resource management systems with varying computing architectures. In other words, even if its installation requires one-time work by HPC administrators, the users have multiple benefits, most notably, neither the necessity of rewriting job scripts for various HPC systems nor the requirement to familiarize themselves with low-level details (e.g. file system capabilities, locations, or available memory and CPUs/cores). On top of that middleware resides a pluggable workflow management system that exploits multiple HPC submission sites by running complex multi-task and distributed workflows. We think it is a novel contribution to the field of glaciological sciences to show how scientific workflow management systems can support science and greatly ease users' work on distributed HPC infrastructures.

We have acknowledged the resource administrators here as they helped us not only in the deployment part, but also the post-deployment efforts, such as site-specific configurations performed once for many users, or application-and-workflow-derived requirements. In our revision of the manuscript, we will states more clearly that UNICORE is a prerequisite, just like the installation of the operating system, the deployed models (Elmer and HiDEM) and their underlying numerical libraries -- all of which require administrative work (just like setting up UNICORE), which however needs to be done only once for many users.
* * *
Reviewer:
In the end, the authors have managed to replace 400 lines of shell script (including job-headers) with a dependency on a high-level workflow management system with some services hosted at yet another supercomputing center. I am not convinced that a thorough clean-up/rewrite of the shell script would not have solved the main problems of the old script with way less effort and overhead.

Reply:
It seems that we need to emphasize the main point of the article more clearly: the abstraction (and hence decoupling from specific platforms) of the workflow. We admit that "a thorough clean-up/rewrite" may alleviate the main problems in a single instance on a single HPC system, however, it would still restrict the execution to this single HPC system the script is tailored to and executed on. In other words, its applicability would remain confined to a static environment. Any change in the setup of the underlying system would inevitably induce changes in the workflow-script and force the user (usually not the administrator) to permanently maintain her workflow code. UNICORE, on the other side, allows to build an abstraction layer on top of the underlying systems (mind the plural). This means that workflows are

   a) Largely decoupled from changes in the operations of the underlying hardware (i.e., HPC cluster, supercomputer) and system software (e.g. job submission system, resource management system specifics);
   b) Capable of running dedicated elements (such as Elmer and HiDEM) on computer systems that are tailored to the task (thereby increasing efficiency), and are specifically optimized and configured by HPC experts of the system once for many users;
   c) Easy to maintain for the user, as the abstraction layer of UNICORE removes a large part of the maintenance effort associated with system updates from the user and shifts it towards system administrators. This helps to avoid error-prone manual script edits by users in time-consuming debug sessions, in order to understand underlying system changes.

We apologize if we missed to communicate these important points in sufficiently detail, and will clarify the manuscript with regard to the main motivation/advantage of deploying UNICORE in lieu of a single-instance script environment.
* * *
Reviewer:

The manuscript itself is somehow entangled between the scientific aspects of the ice dynamical problem, and the technical aspects of the software solution for job control on a set of supercomputers. It lacks a clear focus on either side, as can easily be seen from the mixture of plots that describe the glaciological problem (Figs. 1, 2, 3, 9) and the software solution for abstracting the scripting (Figs. 4-8, with Fig. 7 duplicating the right half of Fig 6). While we are introduced to the basic reasons why one would run the different glaciological models for the different sub-problems, and get some insights on the model grids, and which piece of code solves which sub-problem, there is no interpretation of the results shown in fig 8, or direct glaciological relevance of re-writing the coupling. On the other hand, for the description of how the authors took a shell script and turned it into something more high-level, the details of the glaciological problem are largely irrelevant.

Reply:

Again, we must respectfully disagree. The focus of the manuscript IS the application of abstract workflow management to a glaciological problem (so it focuses on both aspects and not a single one), namely, the one of combining long-term, continuum flow models to short-term, discrete element models that are capable of describing the physics of crack propagation and material failure. Naturally, as space is limited, we cannot elaborate the scientific part in full detail, and rather focus on the completely new aspect of the computational implementation using middleware. The results of the mentioned use-case have been described in detail in a previous publication (Vallot et al., 2018) that we regrettably missed to reference in the initial manuscript, but will include in the revision.
* * *
Reviewer:

I find it hard to image that somebody would successfully use ELMER on a supercomputer without being able to read and manipulate a shell script. The fact that all sub-steps in their new solution are done in shell or python scripts, is ignored. Similarly I don't see a problem with the user having to take care of the code that copies files from one supercomputer to the other, or to adjust a job header for a new computer/queuing system. These tasks are usually adjusted within a day, with individual tasks being on the order of 10 minutes for anybody used to the supercomputer.

Reply:

It is important to note that our case study is not just to "use ELMER", but rather a combination of Elmer and HiDEM, combining two different applications, and also invoking mesh and conversion routines. As a matter of fact, we had a Master's student in computer science working on improving that workflow for his MSc thesis, where his first task was to understand the 400-line

shell script. After a few weeks, that student gave up because it was impossible to understand that shell script independently. Only after explanation by the shell script author, it was possible to improve the script over time. According to the scientific user involved in this endeavor, despite having some Bash shell scripting expertise, UNICORE has proven to be a tremendous improvement for automating the sequential/parallel execution of these scientific workflows. It should be noted that those shell and Python scripts that still remain in our UNICORE-based solution are in fact very short, perform exactly one task each, and are thus modular and easy to understand -- in contrast to the original 400-line shell script.

Furthermore, as the middleware provides an abstract way to combine these models in an application-agnostic way, it may thus open such simulations to the wider glaciology community. This will not only be limited to the Elmer-HiDEM combination, but can be used to integrate other applications as well. This contributes not only to proper reproducibility of science in the light of current discussions on the topic data preservation and data management plans, but also encourages the uptake of our solution for similar research. Further, UNICORE provides a possibility to run across platforms, which would not be possible with a single Bash script running on a front node. The scripts were initially tested on a low-end VM, and could then be run on supercomputers without having to change a single line of code. This is a considerable improvement for scientific communities working on complex problems involving a composite mix of different applications and sharing results in a collaborative manner.
* * *
Reviewer:
Sadly, we never learn, how much resources each of these bits requires and whether there is a point in increasing the complexity of the problem to two supercomputers with remote file transfers for gaining computational efficiency, or whether simply letting the smaller model run in a separate (slightly inefficient) job on the same computer would have been faster in the end.

Reply:

Section 3.3 specifies resource requirements for each workflow task (or job). Naturally, as we just applied changes in the workflow, and the main computation time is consumed by the unaltered models that are coupled therein, the resource requirements have not changed at all. However, we would like to point out that by an improved error handling in the middleware (which is difficult, if not impossible to implement in a Bash script) and the opportunity to exploit system availability across several platforms (hence minimizing queuing times), UNICORE represents a solution with a lot of potential to optimize the overall workflow development and management lifecycle. Admittedly, it is difficult to quantify these gains.

We would also like to point out that -- depending on how "slightly" the inefficiency is interpreted -- some HPC centres demand some minimum code performance, which could render the reviewer's suggestion to run inefficient jobs on the same platform infeasible. Even though running the smaller model on the same computer -- as suggested -- might be faster for the individual scientist, it would result in a sub-optimal resource utilisation. The goal of the HPC community is not only to give individual users their results as fast as possible, but also to utilise the existing hardware as well as possible, to allow running more jobs in the same time. As HPC systems are usually ~99.9% full of jobs on a daily basis, computation time is a precious resource that all scientists should strive to use in the most efficient way.

Finally, regarding file transfers, they can be easily be crafted in scripts if the types of data sources and sinks (http(s), gridftp, scp, sftp...) are known and fixed. But it is unclear how we can manage them if these types are not known in advance while delivering our workflow template to other users, unless we have written a very well-thought-out script. Experience shows that these factors are not always considered when developing scientific applications. Furthermore, the available transfer protocols and their security configurations are changing from HPC site to site and thus can be nicely viewed as another clear benefit of using UNICORE. In UNICORE, we have shown that such parameters can be changed without affecting any scripts.
* * *
Reviewer:
While the authors claim a focus of their work was on improving performance,
(p.10 l26 "focused in particular on improving overall runtime [,. . .]"), they
never provide any information about the results of this endeavor in terms of reduction
of overall runtime, or similar metrics.

Reply:
Thank you for pointing to this potentially misleading statement. Rather than improving performance, we should state more clearly that the major aim of our research is to reduce the overall time and effort required for the workflow design, development, workflow-wide and group-wise iterative and conditional constructs, shared and confined scopes, and monitoring and debugging in a platform independent manner. While we indeed mention runtime on page 10, line 26, this was meant in the context of the attributes just mentioned, and will be clarified in a revision. Furthermore, in Section 8 we explicitly state that "this discussion doesn't cover the application performance", to emphasize which aspects we are focusing on.
* * *
**References**

Vallot, D., J. Åström, T. Zwinger, R. Pettersson, A. Everett, D.I. Benn, A. Luckman, W.J.J. van Pelt, F. Nick, and J. Kohler, 2018. ***Effects of undercutting and sliding on calving: a global approach applied to Kronebreen, Svalbard***. The Cryosphere, **12**, 609-625, [doi:10.5194/tc-12-609-2018](doi:10.5194/tc-12-609-2018)

---

## Referee Comment (RC2) · Anonymous Referee #2 · 18 Sep 2018

General comments

This manuscript describes a new workflow implementations based on the state-of-the-art UNICORE middleware suite to automate and simplify complex task structure across distributed and heterogeneous computing environments. The authors take as a case study an example of coupling two numerical models, one for continuum dynamics of a glacier (Elmer Ice) and a discrete model for calving processes (HiDEM), to demonstrate the progress made by this new implementation. This is an interesting paper that describes a new coupling method that will make the lives of scientists that want to couple different models, even on different platforms, much easier, if the implementation is

as simple as described. The progress in process understanding in many areas, including Earth Sciences, climate research and glaciology, calls for coupling of models, for example climate models with dynamic ice sheet models, ocean circulation models with ice shelves and many others in addition to the example taken in this paper. There will be increasing numbers of coupling tasks in the future and this method opens a way forward in making this kind of task more straight forward and less error prone.

The paper is well written and clear, even though it is somewhat long, the problem is well described, the issues with the initial workflow and the requirement analysis is useful for assessing how well the new method improves the workflow. Authors manage to convince that the new workflow is a great improvement to the previous bash-shell script and will make adaptation of new processes relatively easy. They state that this workflow implementation can be exported for re-running simulations on different platforms, it can be re-used to adapt and apply to new datasets and easily extended. I have only a few minor comments to details in the text, see below.

Specific comments

Title: suggest to put "model" in plural

Page 4, line 1, suggest to find an easier name for "the glacier coupling and calving use case" – and in other places use this name, it will make reading easier

Page 4, line 2, add "a" in front of "part"

Page 4, line 14, suggest to replace "largely" with "extensively"

Page 4 line 24, suggest to edit "the later-on extruded footprint" is not clear

Page 4, line 30 suggest to replace "long-time" with "long-term"

Page 5, line 11, missing r in through

Page 12, line 32, can you state how much "significantly reduced" is?

Page 16, line 17, missing t in "It also..."

Page 18, line 28, missing year in reference

Page 19, line 5, is a repeat, it has already been stated, maybe it is possible to combine these sentences?

Page 23, line 29, suggest to replace "huge" with a quantitative statement

Page 25, line 1, suggest to replace "verify" with "validate" and "glaciology" with "glaciological"
* * *

---

## Author Comment (AC2) · 21 Sep 2018

We are grateful for the reviewer's time and the comments towards improving our manuscript. Our response to the suggested edits and improvements is stated below:
* * *
Page 4, line 1, suggest to find an easier name for "the glacier coupling and calving use case" – and in other places use this name, it will make reading easier

We have now adopted the term "glacio-coupling use case" for this.

[Figure]

Page 12, line 32, can you state how much "significantly reduced" is?

This line is now more streamlined and better describes the main advantages of our approach. It includes the code usability and separation of each of the defragmented workflow steps into a set of interlinked reusable tasks. Furthermore, there are supporting statements being added, that answer: why the workflow code is reusable and what benefit it yields in terms of extensibility and resource heterogeneity.

Page 18, line 28, missing year in reference
These are actually not references, but the code names of the processor microarchitectures produced by Intel, now clarified this in the text.

Title: suggest to put "model" in plural

As both the phases, coupling and calving incorporate one single model separately therefore we think "model" being singular would be more suitable here.

Page 4, line 2, add "a" in front of "part"
Page 4, line 14, suggest to replace "largely" with "extensively"
Page 4 line 24, suggest to edit "the later-on extruded footprint" is not clear
Page 4, line 30 suggest to replace "long-time" with "long-term"
Page 5, line 11, missing r in through
Page 16, line 17, missing t in "It also..."
Page 19, line 5, is a repeat, it has already been stated, maybe it is possible to combine these sentences?
Page 23, line 29, suggest to replace "huge" with a quantitative statement
Page 25, line 1, suggest to replace "verify" with "validate" and "glaciology" with "glaciological"

We appreciate the careful proofreading and have incorporated these corrections and stylistic improvements in the text.

**One minor change:**
In the last manuscript version there was no citation on the actual glacio-coupling use case which describes the underlying scientific significance and methods. The reference is now added to the latest draft on Page 4, line 8-9.

The modified manuscript is attached as a supplement to this reply.

Please also note the supplement to this comment:
https://www.geosci-model-dev-discuss.net/gmd-2018-158/gmd-2018-158-AC2-supplement.pdf

**Supplement:**

[revised manuscript text omitted]

---

## Referee Report (RR1)

The manuscript *Scientific Workflows Applied to the Coupling of a Continuum (Elmer v8.3) and a Discrete Element (HiDEM v1.0) Ice Dynamic Model* has substantially benefited from the last revision. The paper now is much more reader-friendly than before.

I still am not fully convinced by the idea of first listing the steps in the workflow, then stating requirements for the rewrite, then describing the realization of the workflow, and then showing how the new workflow fulfills the requirements. I largely regard this as a stylistic decision, that is left to the authors, but during reading I am left with the impression, that I'm repeatedly reading similar passages. One such case is the overlap between sections 3.3 and 4.3.2. I would suggest to revisit the two sections and perform some streamlining (maybe use exactly the same structure in both, so referencing 3.3 in 4.3.2 is easier?).

Sec 6 Related work is unexpected in this location. The information looks like it could be part of the overview of the state-of-the-art in the introduction. Also, please re-consider which parts of this section are relevant to the reader.

Physics and setups:

Section 3.3 could use a few more references to Vallot et al. (2017,2018) regarding the set-ups. The prescribed temperature field could be briefly explained.

In section 3.3 step 3, it is stated that "HiDEM scales down the obtained friction parameters it receives from Elmer/Ice (in our case using the factor $10^{-4}$) so as to increase the sliding speeds and thereby reduce the physical time (in our case 100 s) needed to evaluate the resulting fractures."
At this point more information on how the stress field is preserved in this scaling and which parameters are affected by the scaling would be very helpful, especially to other scientists planning similar couplings. Is viscosity scaled as well? Any other constants or fields? Is this a feature of HiDEM, or is this done in the transfer scripts? Can you provide evidence of the successful rescaling of the equations?
Also, the reasoning for the separation of time scales permitting the rescaling needs to be more precise than "are separate anyhow" (p7 l30), especially considering the rescaling by a factor of $10^4$ (turning a second into a few hours).
A reference to the prescribed temperature field, that removes the effects of strain heating and basal frictional heating (which would grow drastically with upscaled velocities) might also be helpful.

Specific comments:
page 3 line 5: Gagliardini et al. (2013) needs an *e.g..* This is just one (rather new) ice dynamics model.
Page 3 line 20: *fraction* should probably be *fracturing*.
Page 3 line 28: *very complex* – that's subjective. *complex* should do. Similarly *two (or  more).*
Page 7 line 4: *basal friction law of any type* sounds strange. Maybe just say *a Weertman friction law*
Page 7 line 9: Cuffey and Peterson needs an e.g.

Page 7 lines 19, 21 , also lines 21-23  There is no need to justify that the input data and directories need to be provided before the models can be started.

Page 8 line 21:  The agnostic part is probably requirement 11.

Page 9 line 13/14: Please rephrase.

Page 9 line 24  show me an error-free piece of code…

Page 16 lines 9 … We can consider it standard good practice to bail out of a job when one of the components has failed. That's not a very innovative thing relying on UNICORE, but standard housekeeping.

Page 19 Section 5.4 Lines 5/6 and 9/10 seem to be duplicates. Consider rephrasing the subsection to clarify.

Page 19 lines 19…
"one could argue … workflow." maybe just say that this is facilitated by UNICORE, instead of lamenting about the difficulty of writing a job header.

Page 19 line 33 ""

Page 22, line 9: consider replacing "fire-and-forget pattern" with something that's not missile-terminology.

Figures:
On the way from Vallot 2018 (there Fig. 3) to Memon et al., Fig. 1 gained crevasses in the sketch, but lost "+1" in the bottom index in the right part of the Figure.

Fig 5 still is the right half of Fig 4, and shares a lot of information with Fig. 2. Please change this or make the intentions of this duplication clear in the captions.

---

## Referee Report (RR2)

The clarity and structure of the manuscript of Memon et al has benefitted from the last revision. What remains is a major problem in the (description of the) physics. It is the same problem as in the last revision, where I asked if the viscosity was adjusted with the basal friction parameters.

On the transition from page 7 to page 8 (numbering in the manuscript with change tracking), the authors state

> "The stress field in both models is a consequence of gravity acting on the specific ice geometry and thereby initially identical (differences arise only by discretization method). [...] This allows us to scale down the obtained friction parameters HiDEM receives from Elmer/Ice (in our case using the factor $10^{-4}$) so as to increase the sliding speeds and thereby reduce the physical time (in our case 100 s) needed to evaluate the resulting fractures [...]"

Actually, the stress field results from the balance of gravity and the forces acting on the boundaries, such as basal friction, the support from the bed, and lateral forces. Thus, reducing basal friction generally affects the stress field.
In the HiDEM description paper, Åström et al. (2013),
www.the-cryosphere.net/7/1591/2013/ are aware of the relevance of viscosity when rescaling the equations for acceleration and scale their model parameters accordingly.

Page 1593:
> "Computational problems arise, however, from the fact that the time step length is limited by the rapid timescale of the brittle failure events to approximately $10^{-4}$ s, while the relevant time scale for viscous flow of ice is much longer. To cover both relevant time scales in a single simulation is impractical. It is however possible to use lower viscosities and thereby higher strain rates and re-scale the simulation time to match ice behaviour as long as the viscous flow timescale remains slow compared with that for fracture events (Riikilä et al., 2013). "

Page 1595:
> "The particle model parameters are set such that the resulting viscosity is $10^5$ times lower than in the Elmer model, leading to $10^5$ times faster strain rate, i.e. strain rate is proportional to the inverse of viscosity. "

I recommend involving Jan Åström more closely on this issue, and refer to my comment from the previous iteration:
In section 3.3 step 3, it is stated that
> "HiDEM scales down the obtained friction parameters it receives from Elmer/Ice (in our case using the factor $10^{-4}$) so as to increase the sliding speeds and thereby reduce the physical time (in our case 100 s) needed to evaluate the resulting fractures."

At this point more information on how the stress field is preserved in this scaling and which parameters are affected by the scaling would be very helpful, especially to other scientists planning similar couplings. Is viscosity scaled as well? Any other constants or fields? Is this a

feature of HiDEM, or is this done in the transfer scripts? Can you provide evidence of the successful rescaling of the equations?

Specific comments:

Page 1 line 4, 17 maybe replace "involved models" with "models"
There are quite many unnecessary qualifiers spread over the document. I've listed some below.

Page 1 lines 4 and 5 "and in the worst case even lead to sub-optimal CPU utilization" – maybe just write "and can lead to sub-optimal CPU utilization" and leave it to the reader to define their own worst-case scenario (I'd often put lost simulations higher on my list of nightmares).

P4 line 2 "coupling and on models description" – maybe "coupling and on the models"

Line 31 "were simulated by different models in an offline coupling" – maybe "were simulated by a sequence of different models in a one-way coupling"

P5 line 21
"surface that is  meshed"

P7 line 11 I'd suggest:
" Kronebreen in this part of the workflow"

P8 line 33, I'd suggest:
"the  tasks are well-segregated and do not overlap .

P10 line 8
"the  workflow"

P11 L 4
"the  applications"

P15 L 3
The listing is not to be found in the manuscript.

P20 L 8
"the  models"

P20 L 13/14 I think, this sentence can be removed
"For instance, … already implied here"

---

## Author Response (AR2)

This response describes how our manuscript has been revised to address the recent review comments. We hope to now better describe how the presented work has been realized and how it benefits the ice sheet modelling community in general. The focus is now on highlighting the more efficient CPU utilization, improved I/O management across the workflow tasks, and also the aspect of reproducibility and reusability.

**RC: I still see a serious problem in the lack of focus and coherence of the manuscript. As it currently stands, this manuscript is a mixture of the description of the details of the ice sheet model setup of Vallot et al. (2018) and the usefulness of UNICORE for running models. The latter should be fairly independent of the details of all software steps employed. The glaciological figures still only have minimal importance to the story of the manuscript (and Fig. 2 actually seems to be straight from Vallot et al. (2018) – might want to reference that). The authors should re-consider the figures and their necessity for the main story. For the details of the glaciology we can read the TC paper, now that it is referenced. Then there is a bunch of technical details of the protocols employed inside the UNICORE infrastructure – some of these might be better placed in the UNICORE technical documentation.**

Concerning the use case, or the glaciological coupling, we agree that Figures 1 and 3 can be removed, but Figure 2 - which in fact does not come straight from the TC paper - should remain. The difference is actually crucial as the TC paper does not describe a full coupling, but rather what could be called an offline coupling, which is "only" executing one time step starting from observations (rather than from a model solution of a previous timestep). This is the reason why we believe that some details have to be kept in place for the general understanding of the paper, because this is the first time these two models are coupled together. In this sense, there might have been some misunderstanding regarding the difference between the TC paper and the present one. If a potential user wants to use the full coupling, the present paper is the one to cite and to follow as a guideline. Whereas the present paper describes a full coupling that feeds the output of one model as input to the other, the TC paper is only describing an offline coupling using observations and historical data. This is now described in more detail in the beginning of Section 3.

**RC2: The manuscript needs to focus on the improvements over the commonly accepted good practice, and not on the clean-up effort that was involved in this work. The concept any journal is to present improvements over the state-of-the-art. Of course I am able to fork a run without major effort on my production setups, and my shell scripts give me a decent idea on where they fail – in bash "set -evx" usually does that job. When you have guests, they don't want to know how long it took you to tidy up your place, but what that beautiful piece of art on your wall is. Similarly, I couldn't care less about the problems of the setup before cleanup. So, please feel free to detail on where UNICORE provides an improvement over cleaned-up scripts, but not on the ways, bad scripts are trouble.**

This is certainly an important comment that we took into consideration. Even though there was a previous bash script, we now have focused the paper more on the development of the coupling itself (with the help of UNICORE) rather than on the improvement of the bash script. This has the advantage of being more understandable and interesting for the reader.

We removed the section describing the issues of the bash script as we want to focus on the result and the chosen solution. We rearranged the subsections dealing with the workflow under a common section (4. Workflow). We removed a whole paragraph describing UNICORE details in the implementation section, and also removed too detailed figures and descriptions that are not specific to the use case.

**RC3: As the manuscript is virtually unchanged, it still suffers from serious problems, that prevent publication in its current form. For a revision, the authors need to address those problems and this will require rewriting large parts of the manuscript.**

In the new revision, multiple sections have been rewritten and restructured to better portray our work. Following is a summary of the major changes:

- Abstract, Introduction and Conclusions have been modified to provide more clear and supporting statements on our contributions and the impact of the presented work
- Chapter 4 has been modified; specifically:
    - Section 4.1 (Requirements) has been streamlined with more focused content of relevance to our solution.
    - Section 4.2 now only provides a short introduction to UNICORE. More details related to technical components have been removed.
- Chapter 5 (Discussion) has been overhauled with new subsections to portray our analysis, benefits and enhancements. The sections affected in particular are "Data Transfer and Management", "Efficient Resource Utilization", and "Extendable Workflow Structure", as well as some updates to Sections 6.7 and 6.8.

For a detailed overview of the changes, we refer to the change-tracked PDF difference file gmd-2018-158-diff.pdf that compares the current version to the last version submitted on 21 September 2018 (submitted during the interactive discussion phase).

**RC4: Finally, claims like "the user shouldn't have to be able to handle shell script" or "users shouldn't have to know their supercomputer / their queuing system" really do not help your case. Users who are not able to handle standard shell scripts and job headers are in deep trouble on supercomputers. However, there is no lack of support for learning these skills and quickly solving standard problems. These claims cost you substantial amounts of credibility. Please focus on real improvements your system can bring instead of creating and solving pseudo-problems.**

We respectfully disagree. The justification for constructing the presented workflow is clearly described by the requirements given in the text, which follow from our long-time experiences with supercomputer users. We believe that the reviewer's argument may have been based on a misinterpretation of our text, as the claims described by the reviewer are not to be found within our manuscript. We apologize, should our phrasing of the problem have contributed to this misunderstanding. We hope that our revisions of the text now avoid any such ambiguity.

[revised manuscript text omitted]

---

## Author Response (AR3)

In this response phase we have revised the last submitted manuscript according to the recent review. We are very grateful for your time in reviewing our manuscript and the valuable feedback for improving the presentation of our research.

I still am not fully convinced by the idea of first listing the steps in the workflow, then stating requirements for the rewrite, then describing the realization of the workflow, and then showing how the new workflow fulfills the requirements. I largely regard this as a stylistic decision, that is left to the authors, but during reading I am left with the impression, that I'm repeatedly reading similar passages. One such case is the overlap between sections 3.3 and 4.3.2. I would suggest to revisit the two sections and perform some streamlining (maybe use exactly the same structure in both, so referencing 3.3 in 4.3.2 is easier?).

To avoid any repetitions, the workflow steps described in 4.3.2 of the URC workbench are now briefly summarized (instead of the Listed representation) on Page 14, Lines 3-7.

Sec 6 Related work is unexpected in this location. The information looks like it could be part of the overview of the state-of-the-art in the introduction. Also, please reconsider which parts of this section are relevant to the reader.

For more clarity of our readers, we have now moved the relevant related work discussion into Section 2.2 Scientific Workflows and also labelled Section 2's title as State of the Art.

**Physics and setups:**

Section 3.3 could use a few more references to Vallot et al. (2017,2018) regarding the setups. The prescribed temperature field could be briefly explained.

In section 3.3 step 3, it is stated that "HiDEM scales down the obtained friction parameters it receives from Elmer/Ice (in our case using the factor 10–4) so as to increase the sliding speeds and thereby reduce the physical time (in our case 100 s) needed to evaluate the resulting fractures."

At this point more information on how the stress field is preserved in this scaling and which parameters are affected by the scaling would be very helpful, especially to other scientists planning similar couplings. Is viscosity scaled as well? Any other constants or fields? Is this a feature of HiDEM, or is this done in the transfer scripts? Can you provide evidence of the successful rescaling of the equations?

Also, the reasoning for the separation of time scales permitting the rescaling needs to be more precise than "are separate anyhow" (p7 I30), especially considering the rescaling by a factor of 10  $^4$  (turning a second into a few hours).

A reference to the prescribed temperature field, that removes the effects of strain heating and basal frictional heating (which would grow drastically with upscaled velocities) might also be helpful.

HiDEM is a model for brittle-elastic materials and hence does not take viscous deformations into consideration. Consequently, there is no need for a scaling of the viscosity or any other

parameters. Deformations within HiDEM are solely evaluated using elastic parameters of the ice and - despite calving dynamics - usually negligible in size. Stress fields inside HiDEM - just like in the Stokes problem solved by Elmer/Ice - are rather an instantaneous result to the forces imposed on a certain geometry and independent from the exact timescale, as long as we can be sure that those timescales fall well below those of viscous ice-deformation. This is easily fulfilled, as the fracture time-scale is determined by the ratio of vertical scales and the velocity of sound, and hence is in the sub-second range. Ice flow, on the other hand, happens on the scale of relaxation time, which is dominated by a large viscosity in relation to a significantly smaller Young's modulus, and is thereby in the range of hours and beyond - several orders of magnitude larger. In other words: everything that happens in HiDEM can be interpreted as instantaneous for the ice-flow model, Elmer/Ice. Along the same line of argumentation, HiDEM further does not account for thermodynamic effects, simply because the timescales discussed before are irrelevant to have a significant effect on the global energy balance.

**Specific comments:**

page 3 line 5: Gagliardini et al. (2013) needs an e.g.. This is just one (rather new) ice dynamics model.

The citation has been changed accordingly.

Page 3 line 20: fraction should probably be fracturing.

This has now been changed to "fracturing".

Page 3 line 28: very complex – that's subjective. complex should do. Similarly two (or even more).

This has now been streamlined to "complex as multiple"

**Page 7 line 4: basal friction law of any type sounds strange. Maybe just say a Weertman friction law**

This is mentioned because in our case, any type of friction law (Weertman, Budd, Schoof-Gagliardini, Tsai for example) can be implemented. Currently it is the Weertman friction law. But it is important to know that this can be generalised to some other law. In this case, we remove the "of any type" statement, which might be misleading and would rather state: "basal friction (currently Weertman friction law)."

Page 7 line 9: Cuffey and Paterson needs an e.g.

The citation has been changed accordingly.

Page 10 lines 19, 21 required , also lines 21-23 Without input ... is a necessary prerequisite for all further iterations. There is no need to justify that the input data and directories need to be provided before the models can be started.

"Without [...] further iterations" has now been removed.

Page 8 line 21: in a batch system-agnostic manner. The agnostic part is probably requirement 11.

It is redundant to mention this statement in R2. We have removed this word to avoid replication.

Page 9 line 13/14: Please rephrase.

The description of R6-Parametric Execution was misleading, we have now simplified it to make it more understandable.

Page 9 line 24 against any errors show me an error-free piece of code...

We apologize for the misunderstanding. At this location, we actually meant the run-time errors which may occur due to different reasons, such as system dependent static variables or non-existent data source/sinks. We have rephrased this now to better reflect our point of view.

Page 16 lines 9 ... We can consider it standard good practice to bail out of a job when one of the components has failed. That's not a very innovative thing relying on UNICORE, but standard housekeeping.

Apart from improving the overall workflow makespan, there are also other significant benefits a UNICORE-based implementation offers, such as user intervention while a workflow is in the running state - e.g. halt, develop, restart sequence, use of shared and collaborative cross-site file systems, and access to jobs' working directories. Based on our experience, this is what most scientific users would like to have properly managed while working on different experiments. We have added a corresponding sentence at the end of Section 4.3.3.

Page 19 Section 5.4 Lines 5/6 and 9/10 seem to be duplicates. Consider rephrasing the subsection to clarify.

The duplicate statements have now been removed, and some parts of Section 5.4 have been streamlined for clarity.

Page 19 lines 19...

"one could argue ... workflow." maybe just say that this is facilitated by UNICORE, instead of lamenting about the difficulty of writing a job header.

The text has now been altered (see Page 19, Lines 8-10). It says, "This is avoided by using UNICORE, which provisions each workflow task to have a separate resource requirement specification and also enables different computing platforms (SLURM, PBS, etc.) to be combined into a single workflow. "

Page 19 line 33 "and in a batch system-agnostic manner"

This statement is corrected.

Page 22, line 9: consider replacing "fire-and-forget pattern" with something that's not missile-terminology.

We apologize for the misleading term. It has now been changed to "asynchronously".

Figures:

On the way from Vallot 2018 (there Fig. 3) to Memon et al., Fig. 1 gained crevasses in the sketch, but lost "+1" in the bottom index in the right part of the Figure.

Figure 1 is now presented with clarity.

Fig 5 still is the right half of Fig 4, and shares a lot of information with Fig. 2. Please change this or make the intentions of this duplication clear in the captions.

Figure 4 was just intended to show the general look and feel of the GUI - it is not essential to the paper and hence has been removed (also to shorten the length). The similarities between Fig. 2 and the current Fig. 5 illustrate a strength of the UNICORE implementation of the actual workflow that is visually defined within the GUI. Thus, we prefer to keep both figures, as the first one defines the generic composition of the workflow, and the second shows its implementation within UNICORE.

**Scientific Workflows Applied to the Coupling of a Continuum (Elmer v8.3) and a Discrete Element (HiDEM v1.0) Ice Dynamic Model**

Shahbaz Memon1,4, Dorothée Vallot2, Thomas Zwinger3, Jan Åström3, Helmut Neukirchen4, Morris Riedel1,4, and Matthias Book4

1Jülich Supercomputing Centre, Forschungszentrum Jülich, Leo-Brandt Straße, 52428 Jülich, Germany 2Department of Earth Sciences, Uppsala University, Uppsala, Sweden 3CSC – IT Center for Science Ltd., Espoo, Finland 4Faculty 
[revised manuscript text omitted]

---

## Author Response (AR4)

**Response to the review on "Scientific Workflows Applied to the Coupling of a Continuum (Elmer v8.3) and a Discrete Element (HiDEM v1.0) Ice Dynamic Model" by Shahbaz Memon et al.**

We are grateful to the reviewer and topical editor for providing their constructive reviews and comments towards making our manuscript more accessible to a wider audience. Please find below our responses to the reviewer's comments.
* * *
What remains is a major problem in the (description of the) physics. It is the same problem as in the last revision, where I asked if the viscosity was adjusted with the basal friction parameters.
[...]
At this point more information on how the stress field is preserved in this scaling and which parameters are affected by the scaling would be very helpful, especially to other scientists planning similar couplings. Is viscosity scaled as well? Any other constants or fields? Is this a feature of HiDEM, or is this done in the transfer scripts? Can you provide evidence of the successful rescaling of the equations?

As mentioned in the last version of the manuscript, HiDEM (the particle model) was run elastic-brittle. We now explicitly added the sentence (Page 8, Line 2): "*This means that in the latter only elastic but no viscous deformations are modelled and transfer of viscous rheology parameters from Elmer/Ice to HiDEM can be omitted*". This clarifies that no viscosity is accounted for in this particular setup of HiDEM, so the question of its scaling is obsolete.

Even if friction is artificially reduced, gravity, density and geometry are not changed. This means, for a given glacier geometry (the one transferred from Elmer/Ice to HiDEM), the by the bed balanced forces are the same. Reducing the friction while maintaining its spatial distribution just changes the reaction (i.e. sliding velocity) to this force balance. In other words, geometric changes due to basal sliding will be faster in the particle model and thereby accelerate the computation, leading (as long as we guarantee that inertia effects do not play a significant role) to the same fracture pattern as if constrained with the original basal friction parameters. This is explained by the newly added sentences (Page 8, Line 10): "*We keep the relative distribution of friction parameters and maintain geometry and the value of density as well as gravity. This does not affect the general dynamics, but just*

*accelerates the process that leads to the fraction pattern."* As we also expressed in the text, this acceleration can be justified by the fact that timescales of elastic-brittle dynamics are orders of magnitude smaller compared to the viscous deformation computed in Elmer/Ice.

Specific comments:

Page 1 line 4, 17: maybe replace "involved models" with "models"
There are quite many unnecessary qualifiers spread over the document. I've listed some below.

Page 1 lines 4 and 5: "and in the worst case even lead to sub-optimal CPU utilization" – maybe just write "and can lead to sub-optimal CPU utilization" and leave it to the reader to define their own worst-case scenario (I'd often put lost simulations higher on my list of nightmares).

P4 line 2: "coupling and on models description" – maybe "coupling and on the models"

P4 Line 31: "were simulated by different models in an offline coupling" – maybe "were simulated by a sequence of different models in a one-way coupling"

P5 line 21: "surface that is being meshed"

P7 line 11 I'd suggest: " Kronebreen in this part of the workflow"

P8 line 33, I'd suggest: "the  tasks are well-segregated and do not overlap each other.

P10 line 8: "the  workflow"

P11 L 4: "the  applications"

P20 L 8: "the  models"

P20 L 13/14: I think, this sentence can be removed "For instance, … already implied here"

All of the above changes have now been applied. The changes can be viewed in the PDFDiff document (found under the supplements).

P15 L 3: The listing is not to be found in the manuscript.

The listing was deliberately removed only from the PDFDiff as it was not compatible with the tool that generates the difference document, but it is present in the actual manuscript.

[revised manuscript text omitted]

---

## Author Response (AR5)

**Response to the review on "Scientific Workflows Applied to the Coupling of a Continuum (Elmer v8.3) and a Discrete Element (HiDEM v1.0) Ice Dynamic Model" by Shahbaz Memon et al.**

We are grateful to the topical editor for providing his constructive comments. Please find our responses below:
* * *
P.1 L. 4: overheads -> overhead
P.1 L. 5: period instead of semicolon
P.1 L. 9: allowed users
P.1 L. 19: is increasingly reflected
P.2 L. 23: introduction, a discussion of the state of the art in glacier calving…
P. 20 L. 30: The abstracted workflow allocated only as many…
P. 8 L. 8: Under this assumption, we scale down the friction parameters HiDEM receives…
P. 9 L. 31: parameterize
All of the above changes have now been applied.

P.3 L. 23: I'm not sure what e-Science means here (Earth science? If so, write that)
P. 21 L. 7/14: again, change e-Science to Earth science
To avoid any confusion, we have now used alternative wordings (e.g. "application scenario" and "case study") instead of "e-Science".

P.6 L. 20: what is the contour Cont?
Now it is defined as, ".. the glacier contour (2d boundaries of the glacier), Cont, .."

P. 18 L. 9: workload?
"Workflow" is indeed correct here.

P. 8 L. 13-15: This revised sentence is still confusing
We have corrected the sentence structure now.

P. 6 L. 8: time steps?
This has been changed to "time-step size".

All the changes can be viewed in the PDFDiff document (found under the supplements).